# Stem cell factor is selectively secreted by arterial endothelial cells in bone marrow

Chunliang Xu[1,2], Xin Gao[1,2], Qiaozhi Wei[1,2], Fumio Nakahara [1,2], Samuel E. Zimmerman[3,4], Jessica Mar[3,4] & Paul S. Frenette [1,2,5]

Endothelial cells (ECs) contribute to haematopoietic stem cell (HSC) maintenance in bone marrow, but the differential contributions of EC subtypes remain unknown, owing to the lack of methods to separate with high purity arterial endothelial cells (AECs) from sinusoidal endothelial cells (SECs). Here we show that the combination of podoplanin (PDPN) and Sca-1 expression distinguishes AECs (CD45$^-$ Ter119$^-$ Sca-1$^{bright}$ PDPN$^-$) from SECs (CD45$^-$ Ter119$^-$ Sca-1$^{dim}$ PDPN$^+$). PDPN can be substituted for antibodies against the adhesion molecules ICAM1 or E-selectin. Unexpectedly, prospective isolation reveals that AECs secrete nearly all detectable EC-derived stem cell factors (SCF). Genetic deletion of *Scf* in AECs, but not SECs, significantly reduced functional HSCs. Lineage-tracing analyses suggest that AECs and SECs self-regenerate independently after severe genotoxic insults, indicating the persistence of, and recovery from, radio-resistant pre-specified EC precursors. AEC-derived SCF also promotes HSC recovery after myeloablation. These results thus uncover heterogeneity in the contribution of ECs in stem cell niches.

---

[1] The Ruth L. and David S. Gottesman Institute for Stem Cell and Regenerative Medicine Research, Albert Einstein College of Medicine, New York, NY 10461, USA. [2] Department of Cell Biology, Albert Einstein College of Medicine, New York, NY 10461, USA. [3] Department of Systems and Computational Biology, Albert Einstein College of Medicine, New York, NY 10461, USA. [4] Department of Epidemiology and Population Health, Albert Einstein College of Medicine, New York, NY 10461, USA. [5] Department of Medicine, Albert Einstein College of Medicine, New York, NY 10461, USA. Correspondence and requests for materials should be addressed to P.S.F. (email: paul.frenette@einstein.yu.edu)

Haematopoietic stem cells (HSCs), at the top of the haematopoietic cell hierarchy, give rise to all mature haematopoietic cells throughout life. HSCs are thought to be maintained in specific niches, allowing their maintenance and regulation of their fate[1–3]. Staining of endogenous HSCs using CD150, CD48, CD41, and lineage expression has revealed that they are broadly distributed close to sinusoidal endothelial cells (SECs)[4]. Subsequent studies have revealed that perivascular stromal cells enriched in mesenchymal stem cell (MSC) activity, marked by *Nestin*-GFP, expressed high levels of the major HSC niche factors and were associated with HSCs in bone marrow (BM)[5]. Selective diphtheria toxin-induced depletion of these cells prevented homing of progenitors in BM[5], whereas the depletion of overlapping CXCL12-abundant reticular (CAR) cells reduced HSC numbers in BM[6]. BM HSCs were also reduced after conditional deletion of *Cxcl12* or stem cell factor (*Scf*) in perivascular cells[7–10]. *Nestin*-GFP[+] stromal cells are heterogeneous in that the NG2[+] peri-arteriolar fraction was found to regulate HSC proliferation[11], and contribute to BM CXCL12 content[10]. Constitutive deletion of *Scf* in endothelial cells (ECs) also reduced HSC numbers in BM, suggesting that ECs contribute to the HSC niche. Co-deletion of *Scf* in perivascular cells (*Lepr*-Cre) and ECs (*Tie2*-Cre) further reduced HSC numbers, suggesting additive and multicellular contributions in HSC niches[7].

Whether SCF is homogenously secreted by ECs in the BM remains unclear because there is thus far no method to separate precisely EC subtypes[12]. Endomucin[+] CD31[Hi] (Type H) ECs have been identified, near the growth plate and endosteum of young mice, in vessels bridging sinusoids and arterioles where they regulate osteogenesis in a Notch-dependent manner[13–15]. Sca-1 antigen has been used to identify arterioles in imaging analyses of the BM[11,16], but whether Sca-1 can be used by itself to isolate AECs by fluorescence-activated cell sorting (FACS) is unknown.

Here, we report that the combination of podoplanin (PDPN) and Sca-1 expression can prospectively distinguish BM AECs from BM SECs. We show that EC-derived SCF is exclusively produced by the arterial vasculature of the BM and that selective *Scf* deletion in AECs, but not SECs, alters BM HSC numbers. AEC-derived SCF also promotes HSC recovery after myeloablation. Furthermore, we demonstrate using lineage tracing that the regeneration of the BM vasculature after myeloablation, is accomplished independently by pre-specified arterial and sinusoidal radio-resistant precursors.

## Results

**Separation of arterial and sinusoid ECs with PDPN and Sca-1.** BM ECs are commonly identified as CD31-expressing cells among the non-haematopoietic CD45[−] Ter119[−] fraction. Sca-1 expression was previously shown to mark the arterial vasculature by confocal immunofluorescence analyses of the BM[11,16]. To evaluate the ability of Sca-1 expression to isolate prospectively arterial endothelial cells (AECs), we stained flushed BM nucleated cells with antibodies against CD45, Ter119, CD31, and Sca-1. FACS analyses revealed that the vast majority (~80%) of CD45[−] Ter119[−] CD31[+] cells co-expressed Sca-1 (Supplementary Fig. 1a), suggesting that Sca-1 expression was not restricted to AECs, which should comprise a minor fraction of total BM ECs[11]. In vivo injection of antibodies to physiologically labelled ECs (anti-CD31, anti-VE-cadherin, and anti-Sca-1) revealed, by contrast, that virtually all CD31[+] VE-cadherin (CD144)[+] ECs (~99.4%) expressed Sca-1 (Fig. 1a). Whole-mount immunofluorescence analysis of sternal bones of the same mice revealed uniform labelling of the entire vascular network and the higher staining of arteries by anti-Sca-1, suggesting that AECs may be

Sca-1[bright] but cannot be cleanly separated by FACS because SECs also express Sca-1 (Fig. 1b). The difference in staining patterns for Sca-1 between the classical in vitro or the physiological in vivo staining methods implies that a sizable fraction (~20%; compare Supplementary Fig. 1a and Fig. 1a) of in vitro-stained CD31[+] cells are not bona fide ECs.

Previous studies have suggested that the lymphatic vessel marker VEGFR3 can label sinusoids by immunofluorescence imaging[17], but commercial anti-VEGFR3 antibodies cannot distinguish SECs from AECs by FACS. Searching to fractionate the Sca-1[+] ECs, we used another lymphatic EC marker, PDPN[18], which separated CD45[−] Ter119[−] CD31[+] cells into three distinct populations: Sca-1[bright] PDPN[−], PDPN[+] Sca-1[dim], and PDPN[−] Sca-1[−] double-negative cells (Fig. 1c). PDPN is mucin-type transmembrane glycoprotein expressed on lymphatic ECs and fibroblastic reticular cells (FRCs)[19]. To evaluate the spatial localisation of PDPN[+] vessels in the BM, we injected *Nestin*-GFP mice (which brightly marks arteries) with fluorescently-labelled antibodies against PDPN, CD31, and VE-cadherin and imaged whole-mount sternal BM. We found that anti-PDPN, but not an isotype-matched antibody, exclusively labelled SECs, and spared *Nestin*-GFP[bright] arteries (Fig. 1d and Supplementary Fig. 1b). These results suggest that Sca-1[bright] PDPN[−] cells are AECs and PDPN[+] Sca-1[dim] cells are SECs, whereas PDPN[−] Sca-1[−] double-negative cells (in the CD31[+] fraction) may not be ECs.

To evaluate further this issue, we stained ECs in vivo by injection of fluorescently-labelled Isolectin B4 (lectin from *Griffonia simplicifolia*) and anti-VE-cadherin. Both methods similarly stained the vascular network following intravenous injection (i.v.) without staining of the BM parenchyma (Supplementary Fig. 1d). We then evaluated by FACS analysis PDPN and Sca-1 expression on Isolectin B4[+] VE-cadherin[+] ECs extracted from flushed enzymatically dissociated BM. Sca-1[bright] PDPN[−] cells represented 8–9% of ECs identified by Isolectin B4 or VE-cadherin, whereas ~86% of ECs were in the PDPN[+] Sca-1[dim] gate (Fig. 1e). In vivo staining with either Isolectin or VE-cadherin did not reveal PDPN[−] Sca-1[−] double-negative cells, suggesting that all BM ECs indeed express either PDPN or/and Sca-1. To confirm further this issue, we stained enzymatically dissociated BM with antibodies against CD45, Ter119, CD31, PDPN, and Sca-1 from mice that had been injected intravenously with fluorescently-labelled Isolectin and anti-VE-cadherin. Upon gating on CD45[−] Ter119[−] CD31[+] cells, we found that PDPN[−] Sca-1[−] double-negative cells were not labelled by i.v. injected endothelial markers, whereas virtually all AECs (Sca-1[bright] PDPN[−]) and SECs (PDPN[+] Sca-1[dim]) were labelled (Fig. 1f). The majority of SECs (PDPN[+] Sca-1[dim]), but not AECs (Sca-1[bright] PDPN[−]), expressed EPHB4, a receptor tyrosine kinase recently described on BM SECs by immunofluorescence staining (Supplementary Fig. 1c)[20]. These results confirm that the ex vivo-stained CD31[+] fraction contains non-ECs and suggest that PDPN and Sca-1 can be used to study prospectively purified populations of AECs and SECs.

**AECs express high levels of canonical niche factors.** In *Nestin*-GFP transgenic mice, AECs can be further separated as *Nestin*-GFP[bright] and *Nestin*-GFP[dim] fractions, whereas SECs were uniformly *Nestin*-GFP[dim] (Fig. 2a). We next isolated these three populations of ECs (*Nestin*-GFP[bright] AECs, *Nestin*-GFP[dim] AECs, and SECs) to interrogate their transcriptome by RNA-seq analysis. Principal component analysis revealed that *Nestin*-GFP[bright] AECs and *Nestin*-GFP[dim] AECs clustered together, away from SECs (Fig. 2b). Hierarchical clustering analysis of differentially expressed genes also showed the distinct gene expression profile of AECs (both *Nestin*-GFP[bright] AECs and

*Nestin*-GFP$^{dim}$ AECs) compared to SECs, while *Nestin*-GFP$^{bright}$ AECs and *Nestin*-GFP$^{dim}$ AECs showed few genes that were differentially expressed (Fig. 2c). Comparison of the expression profile of AECs and SECs with perivascular stromal cells (labelled by *Lepr*-Cre, *NG2*-Cre, or *Myh11*-Cre$^{ER}$) or i.v.-stained CD31$^+$ cells[10], revealed separate clustering of AECs, SECs, and stromal cell types while total CD31$^+$ (i.v. injected antibody), as expected, were between AECs and SECs (Supplementary Fig. 2). Importantly, RNA-seq confirmed that *Sca-1* (encoded by *Ly6a*) was expressed by both AECs and SECs, and that its expression was considerably higher in AECs (RPKMs in *Nestin*-GFP$^{bright}$ AECs, *Nestin*-GFP$^{dim}$ AECs, and SECs: 2455 ± 1302, 2163 ± 780, and 362 ± 197, respectively), which is consistent with our imaging and FACS analysis results. AECs were highly enriched for expression

of arterial-associated genes, including *Efnb2*, *Gja5*, *Gja4*, and *Bmx* than SECs (Fig. 2d). The higher expression of *Bmx* and *Efnb2* in AECs compared to SECs was confirmed independently using qPCR analysis (Fig. 2e, f). On the other hand, SECs were highly enriched for the expression of the liver SECs genes *Stab2*, *Dnase1l3*, and *C1qtnf1*[21], as well as adhesion molecules mediating haematopoietic stem and progenitor cell homing[22], *Sele*, *Selp*, *Icam1*, *Vcam1* compared to AECs (Fig. 2g). These data validate the identity of AECs and SECs, and uncover their precise gene signature (Fig. 2d, g).

Surprisingly, we found that AECs expressed much higher levels of the major HSC niche factors *Scf* (encoded by *Kitl*) and *Cxcl12* (Fig. 2h and Supplementary Fig. 3). This was confirmed by qPCR analysis of sorted AECs and SECs (Fig. 2i, j). To obtain insight

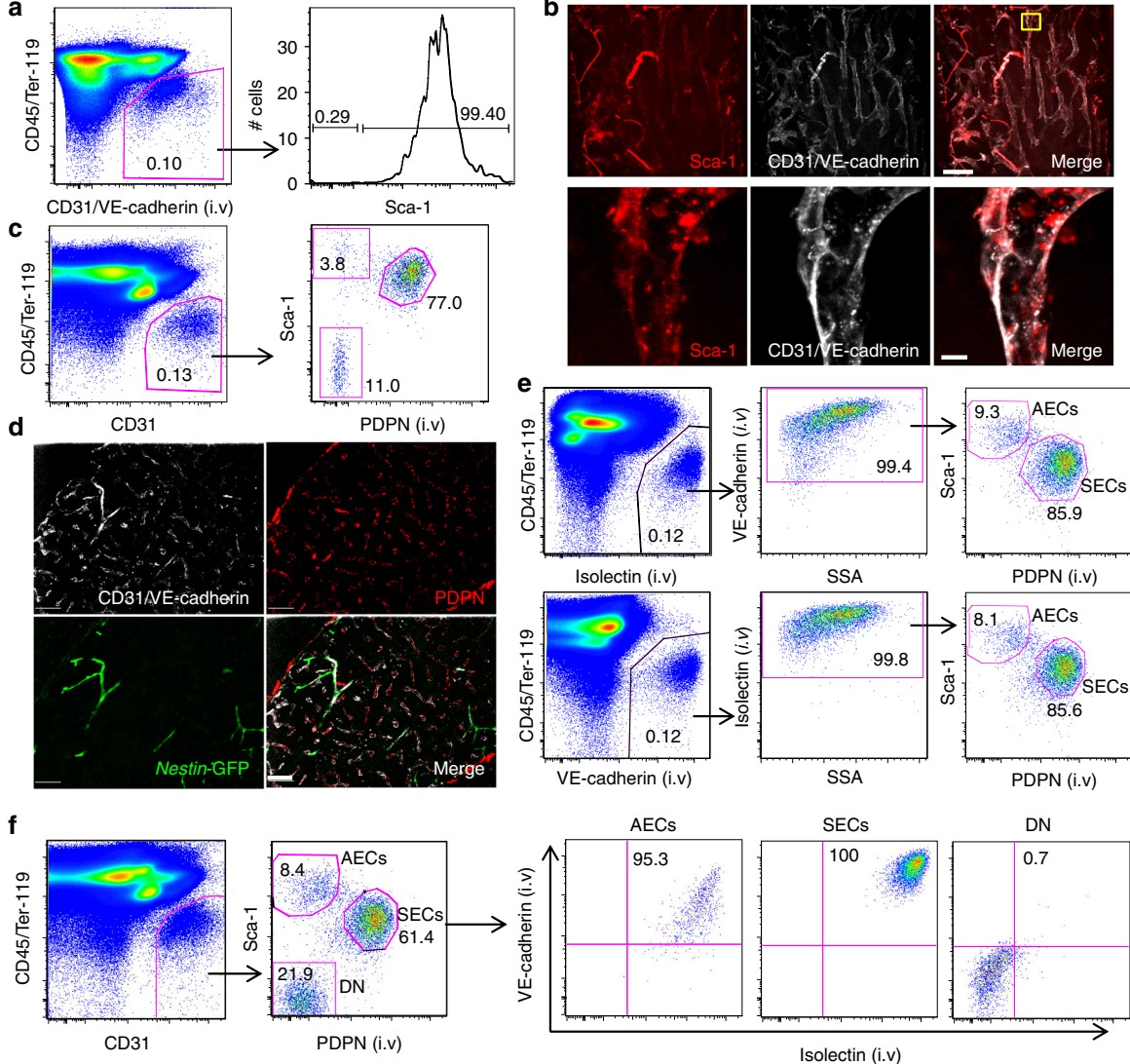

**Fig. 1** Separation of arterial from sinusoidal bone marrow endothelial cells using PDPN and Sca-1 expression. **a** Representative FACS plot of the Sca-1 expression on ECs from mice injected i.v. with anti-CD31/anti-VE-cadherin showing that all ECs are Sca-1$^+$. Cells were pre-gated on singlet, live cells. **b** Representative whole-mount image of sternum from mice treated as in (**a**). Scale bar, 10 μm. **c** PDPN and Sca-1 separate CD45$^-$ Ter119$^-$ CD31$^+$ cells into three populations: Sca-1$^{high}$ PDPN$^-$, PDPN$^+$ Sca1$^{dim}$, and Sca-1$^-$ PDPN$^-$ double-negative populations. Cells were pre-gated on singlet, live cells. **d** Representative imaging of femur BM from *Nestin*-GFP mice stained with anti-VE-cadherin and anti-PDPN by i.v. injection showing that PDPN labels exclusively *Nestin*-GFP$^-$ sinusoids. Scale bar, 100 μm. **e** Representative FACS plot of ECs from a wild-type mouse treated with fluorescently-labelled anti-VE-cadherin and Isolectin GS-IB4 i.v. to label ECs. Virtually all the ECs are within the AECs and SECs compartments. Cells were pre-gated on singlet, live cells. **f** Representative FACS plot of ECs from the same animals as in (**e**). All cells within the AECs and SECs gates are VE-cadherin$^+$ Isolectin GS-IB4$^+$ ECs. Double-negative cells are VE-cadherin$^-$ Isolectin GS-IB4$^-$. Cells were pre-gated on singlet, live cells

into their expression at the single cell level, we analysed ECs in *Scf*-GFP and *Cxcl12*-GFP reporter mice[7,23]. In these genetically engineered animals, the GFP sequence was knocked into the niche gene locus to reflect the promoter activity. Remarkably, the vast majority of AECs were *Scf*-GFP[+] whereas virtually all SECs were *Scf*-GFP[−] (Fig. 2k). Additionally, *Cxcl12*-GFP was expressed at higher level in AECs than SECs (Fig. 2l), and several other known niche factors (e.g. *Jag1*, *Jag2*, *Tgfb2*, *Il6*) were expressed at

higher levels in AECs, compared to SECs (Fig. 2h). These results suggest that AECs may play a more significant role than SECs in HSC maintenance.

Consistent with the higher expression of adhesion molecules on SECs revealed by RNAseq analysis, the higher expression of *Sele*, *Selp*, *Icam1*, *Vcam1* in SECs compared to AECs was also confirmed using qPCR (Fig. 3a), leading us to evaluate whether they could substitute for PDPN. Indeed, we found that antibodies

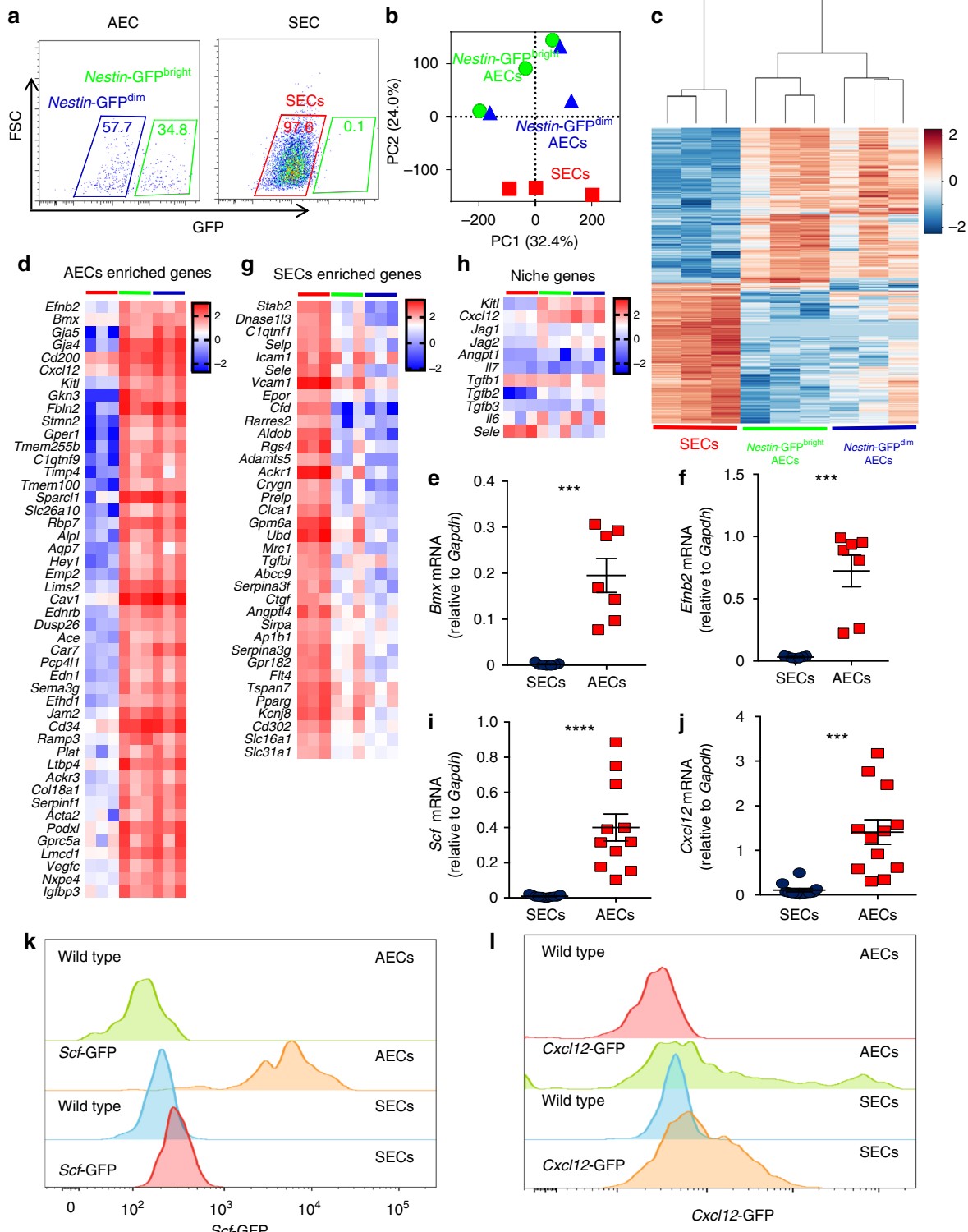

against either ICAM-1 or E-selectin (CD62E) marked all SECs by FACS analysis (Fig. 3b). In addition, anti-ICAM-1 or anti-CD62E antibodies along with anti-Sca-1 could be used to stain all ECs without the need for intravenous antibody injection (Fig. 3c). Furthermore, we have confirmed the high expression of adhesion molecules P-selectin, E-selectin, and ICAM-1 on SECs by immunofluorescence microscopy analysis (Fig. 3d–f). Thus, these results indicate that BM AECs and SECs can be identified or isolated at high purity by ex vivo staining using only four antibodies against CD45, Ter-119, Sca-1, and ICAM-1 or E-selectin.

**SEC-derived SCF is dispensable for HSC maintenance.** Among the differentially expressed genes, we noted that the erythropoietin receptor (*Epor*) was selectively expressed on SECs (Fig. 2g). Although *Epor* expression is well described on erythroid lineage cells[24] and was recently shown to be expressed by BM ECs[25], its specific expression on SECs has, to our knowledge, not been previously reported. qPCR analyses from sorted AECs and SECs confirmed that SECs expressed significantly higher *Epor* gene transcripts than AECs (Supplementary Fig. 4a). Imaging of femoral BM from *Epor*-Cre transgenic mice[26] intercrossed with ROSA26-loxP-stop-loxP-tdTomato (iTdTomato) reporter and *Nestin*-GFP mice (to mark arteries) revealed that SECs were targeted by *Epor*-Cre, while the arterial tree was mostly spared (Fig. 4a, b and Supplementary Fig. 4b). Among BM haematopoietic cells, labelling by *Epor*-Cre was largely restricted to megakaryocyte-erythroid progenitors (MEP; Supplementary Fig. 5), but these haematopoietic progenitors and their progeny were not visible in our immunofluorescence analyses due to the short exposure imposed by the strong iTdTomato signal in ECs. FACS analysis of ECs from these mice showed that *Epor*-Cre labelled $80.0 \pm 2.1\%$ of SECs and a minor proportion of AECs ($21.0 \pm 1.1\%$; Fig. 4c, d). Importantly, *Epor*-Cre did not target CD45[−] Ter-119[−] CD31[−] *Nestin*-GFP[+] MSCs (Fig. 4e). Thus, *Epor*-Cre selectively targets SECs in the BM microenvironment.

To assess the contribution of SEC-derived SCF in the maintenance of HSCs, we intercrossed *Epor*-Cre mice with *Scf*[fl/−] mice and evaluated its impact on the number and function of HSCs and progenitors. Deletion of *Scf* using *Epor*-Cre did not alter *Scf* expression by AECs or MSCs (MSCs: CD51[+] PDGFRα[+] CD31[−] Ter-119[−]; Supplementary Fig. 6a). We found no significant difference in the percentage or absolute numbers of CD150[+] CD48[−] LSK HSCs in the BM between *Epor*-Cre (−) *Scf*[fl/−] and *Epor*-Cre (+) *Scf*[fl/−] littermates (Fig. 4f, g). Competitive repopulation experiments also showed that equal repopulation capacity of BM cells obtained from *Epor*-Cre (−) *Scf*[fl/−] and *Epor*-Cre (+) *Scf*[fl/−] mice (Fig. 3h–k). In addition, we did not observe any significant difference in total BM cellularity or progenitor counts, including Lin[−], Sca-1[+], c-kit[+] cells (LSK), multipotent progenitor cells (MPP), MEP, lymphoid-primed multipotent progenitor (LMPP), common myeloid progenitors (CMP), common lymphoid progenitor (CLP), and granulocyte-macrophage progenitor (GMP) (Supplementary Fig. 6b and c). These data, together with the expression results, clearly show that the SCF derived from SECs is dispensable for HSC maintenance.

**AEC-derived SCF regulates HSC maintenance.** As *Bmx*, a member of the Tec tyrosine kinase family[27], was selectively expressed by BM AECs in our RNAseq and qPCR analyses (Fig. 2d, e), we crossed *Bmx*-Cre[ERT2] transgenic mice[28] with the iTdtomato reporter and *Nestin*-GFP mice to evaluate whether *Bmx*-Cre[ERT2] can target arteries. Femoral BM immunofluorescence microscopy imaging of triple-transgenic mice revealed that *Bmx*-Cre[ERT2] exclusively labelled *Nestin*-GFP[bright] arteries, but not SECs (Fig. 5a, b; Supplementary Fig. 7). We also injected anti-Sca-1 i.v. into *Bmx*-Cre[ERT2]; iTdtomato mice to stain AECs and again found that the *Bmx*-Cre[ERT2] exclusively labelled Sca-1[bright] arteries (Fig. 5c). FACS analysis of BM extracts of these mice revealed that $62.7 \pm 0.7\%$ of AECs were targeted by *Bmx*-Cre[ERT2], whereas SECs were not labelled ($0.01 \pm 0.03\%$; Fig. 5d, e). In addition, CD45[−] Ter119[−] CD31[−] *Nestin*-GFP[+] MSCs were not targeted by *Bmx*-Cre[ERT2] (Fig. 5f). Thus, *Bmx*-Cre[ERT2] can specifically label BM AECs, but not SECs or MSCs.

To investigate the contribution of AECs in SCF secretion, we bred *Bmx*-Cre[ERT2] mice with *Scf*[fl/−] animals. We found that *Scf* deletion using *Bmx*-Cre[ERT2] significantly reduced *Scf* expression in AECs, while it did not alter *Scf* expression in MSCs (Supplementary Fig. 8a). Arterial *Scf* deletion also did not alter EC numbers or the vascular structure of the BM (Supplementary Fig. 8b and c). FACS analysis of BM mononuclear cells showed that CD150[+] CD48[−] LSK HSCs were significantly reduced both in percentage and absolute numbers in the BM of *Bmx*-Cre[ERT2] (+) *Scf*[fl/−] mice compared to *Bmx*-Cre[ERT2] (−) *Scf*[fl/−] littermate controls (Fig. 5g, h). LSK cells were similarly reduced (Fig. 5i, j). Competitive repopulation assay revealed a significant tri-lineage reduction in reconstitution of BM cells from *Bmx*-Cre[ERT2] (+) *Scf*[fl/−] mice compared to cells from control *Bmx*-Cre[ERT2] (−) *Scf*[fl/−] animals (Fig. 5k–n). There was no significant difference in the total BM cellularity, numbers of MEP, MPP, LMPP, GMP, CMP, and CLP in the *Bmx*-Cre[ERT2] (+) *Scf*[fl/−] mice compared to littermate controls (Supplementary Fig. 8d and e). Furthermore, *Scf* deletion by *Bmx*-Cre[ERT2] did not change the cell cycle status of HSCs or the HSC localisation relative to arterioles (Supplementary Fig. 8f and g). These data, together with the expression profile, clearly indicate that BM EC SCF is derived from the arterial vascular tree.

**AECs self-regenerate and do not regenerate SECs.** The origin of ECs during the BM regeneration remains unclear. For example, following irradiation SECs are heavily damaged whereas arteries are thought to be more structurally resilient. Type H ECs were

**Fig. 2** AECs, but not SECs, express high levels of canonical niche factors. **a** Representative FACS plot of AECs and SECs from *Nestin*-GFP mice. SECs, *Nestin*-GFP[bright] AECs and *Nestin*-GFP[dim] AECs were sorted for RNA sequencing. **b** Principal component analysis of gene expression, showing that *Nestin*-GFP[bright] AECs and *Nestin*-GFP[dim] AECs cluster together, while SECs occupy a space different from AECs. **c** Hierarchical clustering of significantly differentially expressed genes. The values shown are z-scores. **d** Heatmap of the genes specific in AECs. Groups are indicated by the colour of the bar on top of the graph as used in panels (**a**), (**b**), and (**c**). The values are log10 transformed-RPKM. **e** qPCR was performed on sorted AECs and SECs to quantify mRNA levels of *Bmx*. $n = 7$ for AECs and SECs. **f** qPCR was performed on sorted AECs and SECs to quantify mRNA levels of *Efnb2*. $n = 7$ for AECs and SECs. **g** Heatmap of genes specific in SECs. Groups are indicated by the colour of the bar on top of the graph as used in panel c. The values are log10 transformed-RPKM. **h** Heatmap of niche genes. Different groups are indicated by the colour of the bar on top of the graph as used in panels (**a**), (**b**), and (**c**). The values are log10 transformed-RPKM. **i** qPCR was performed on sorted AECs and SECs to quantify mRNA levels of *Scf*. $n = 11$ for AECs and SECs. **j** qPCR was performed on sorted AECs and SECs to quantify mRNA levels of *Cxcl12*. $n = 12$ for AECs and SECs. **k** Representative FACS plot of *Scf*-GFP reporter activity in AECs and SECs. **l** Representative FACS plot of *Cxcl12*-GFP reporter activity in AECs and SECs. **e, f, i, j** Data are represented as mean ± SEM. Data were analysed with two-tailed *t*-test. $*p < 0.05$, $**p < 0.01$, $***p < 0.001$, $****p < 0.0001$

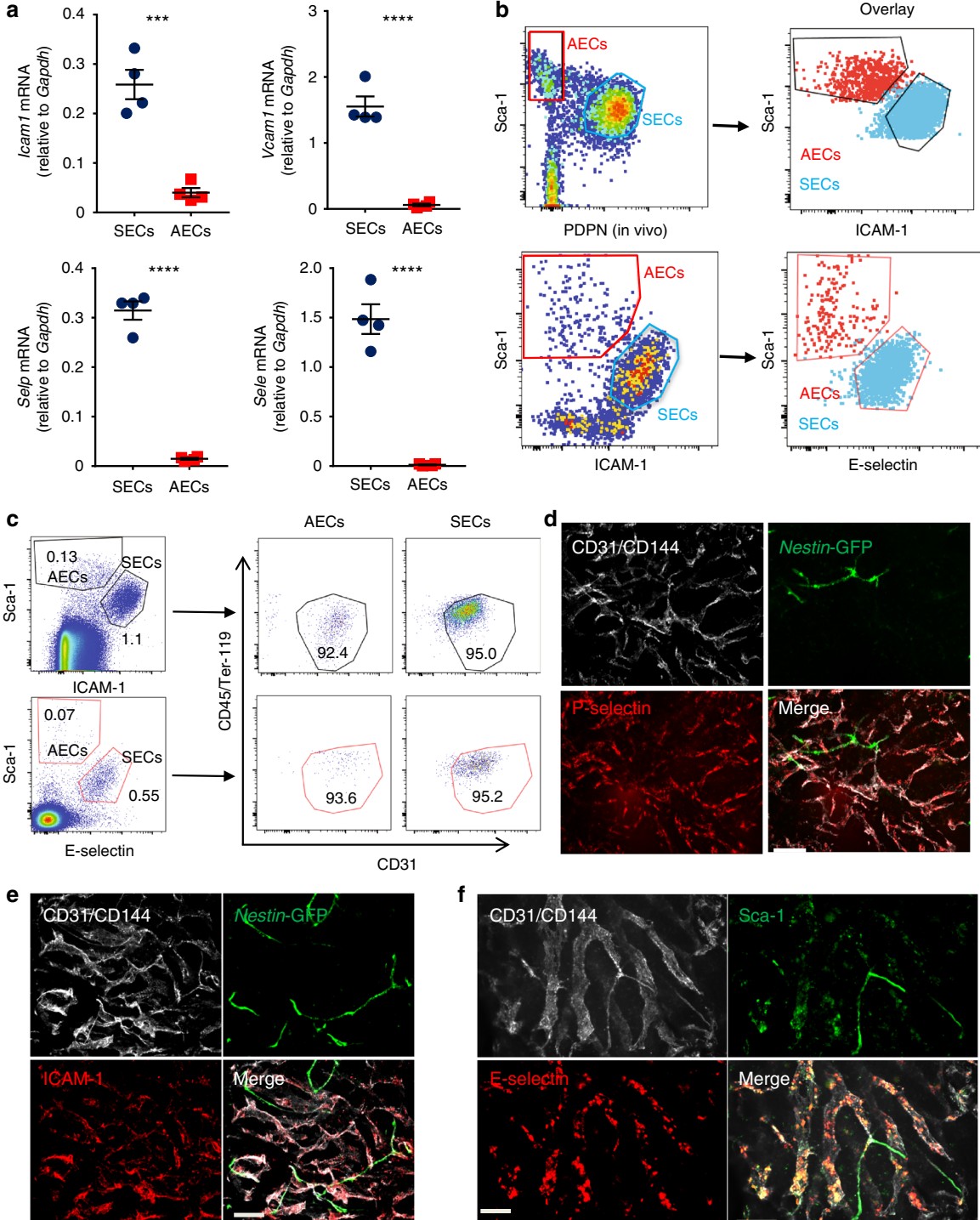

**Fig. 3** Elevated cell adhesion molecule expression on SECs. **a** qPCR analysis of adhesion molecules, *Selp*, *Sele*, *Icam1*, and *Vcam1* of sorted AECs and SECs. Data are represented as mean ± SEM. $n = 4$ for AECs and SECs. Data were analysed with two-tailed *t*-test. *$p < 0.05$, **$p < 0.01$, ***$p < 0.001$, ****$p < 0.0001$. **b** PDPN can be substituted for antibody against the adhesion molecule ICAM1 and E-selectin. Shown is EC isolation from a wild-type mouse injected i.v. with anti-PDPN and the enzymatically dissociated bone marrow cells were stained with other antibodies ex vivo and pre-gated on singlet, live, CD45⁻, Ter119⁻, CD31⁺ cells. **c** Anti-CD31 is dispensable for the analysis of bone marrow AECs and SECs. Flushed enzymatically dissociated bone marrow cells were stained with anti-CD45, anti-Ter119, anti-ICAM-1/anti-CD62E, anti-CD31, and anti-Sca-1 ex vivo. Analysis was pre-gated on singlet, live, CD45⁻, Ter119⁻ cells. **d** P-selectin marks sinusoids. Anti-P-selectin and anti-CD31/CD144 were i.v. injected into *Nestin*-GFP mice and femur was imaged after removal. Scale bar, 50 um. **e** ICAM-1 marks sinusoids. Anti-ICAM-1 and anti-CD31/CD144 were i.v. injected into *Nestin*-GFP mice and femur was imaged after harvest. Scale bar, 100 um. **f** E-selectin marks sinusoids. Anti-E-selectin, anti-Sca-1, and anti-CD31/CD144 were i.v. injected into wild-type mice and femur was imaged after harvest. Scale bar, 50 um

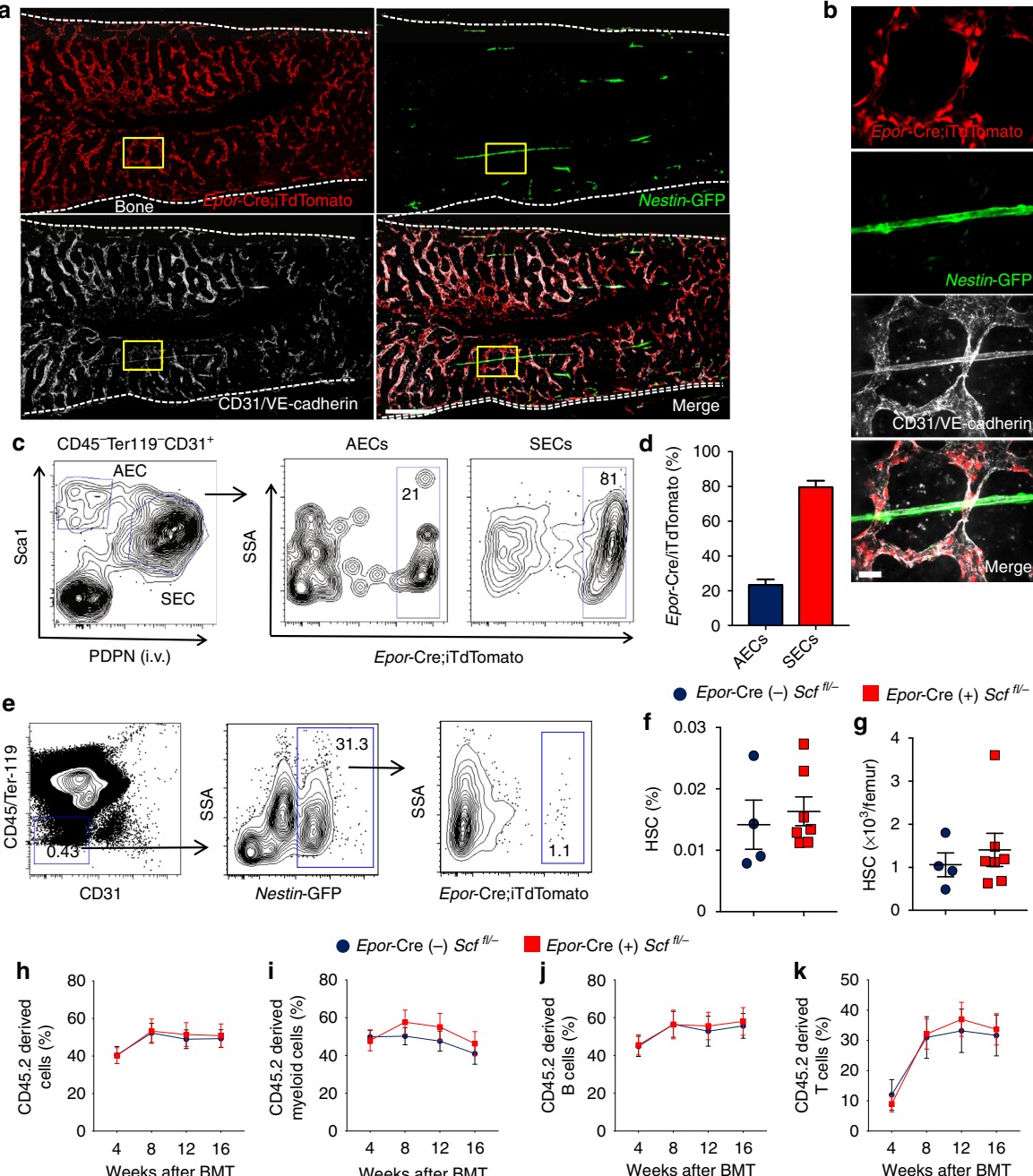

**Fig. 4** SCF secreted by SECs is dispensable for HSC maintenance. **a** Images of femur from *Epor*-Cre;iTdtomato;*Nestin*-GFP mice stained i.v. with anti-VE-cadherin and anti-CD31. All the panels show the same field for different channels. Scale bar, 100 μm. **b** Enlarged images of the field highlighted by yellow rectangles in (**a**). All panels show the same field for different channels. The *Epor*-Cre-labelled cells do not colocalise with *Nestin*-GFP^bright arteries. Scale bar, 10 μm. **c** Representative FACS plot of the labelling of AECs and SECs by *Epor*-Cre. **d** Quantification of the labelling efficiency of AECs and SECs by *Epor*-Cre. Data are represented as mean ± SEM. $n = 7$ for AECs and SECs. **e** Representative FACS plot of *Nestin*-GFP$^+$ MSCs labelling by *Epor*-Cre. **f** Percentage of HSCs (CD150$^+$ CD48$^-$ LSK) in the BM. Data are represented as mean ± SEM. $n = 4$ for Cre (−) and $n = 7$ for Cre (+). **g** Number of HSCs (CD150$^+$ CD48$^-$ LSK) in the BM. Data are represented as mean ± SEM. $n = 4$ for Cre (−) and $n = 7$ for Cre (+). **h–k** Percentage of donor-derived total cells and tri-lineage (myeloid, B and T) cells after competitive transplantation. $n = 7$ for Cre (−), and $n = 3$ for Cre (+). The label of X-axis indicates the time (weeks) after transplantation

reported to give rise to other ECs during the juvenile stage, but these results were based on the differential labelling of ECs in *Cdh5*-Cre^ER mice[13]. We thus evaluated whether regeneration of SECs was dependent on arterial-derived endothelial precursors. *Bmx*-Cre^ERT2;iTdtomato mice were treated with tamoxifen 4 weeks prior to myeloablation[28]. In the first set of experiments,

we treated mice with fluorouracil (5-FU) and monitored endothelial recovery (Fig. 6a). 5-FU treatment significantly reduced the number of SECs and AECs (Fig. 6b). Both AECs and SECs robustly proliferated on day 9 after 5-FU challenge, whereas by contrast, most ECs were quiescent in the steady state (Fig. 6c). FACS analysis of BM ECs revealed that *Bmx*-labelled ECs

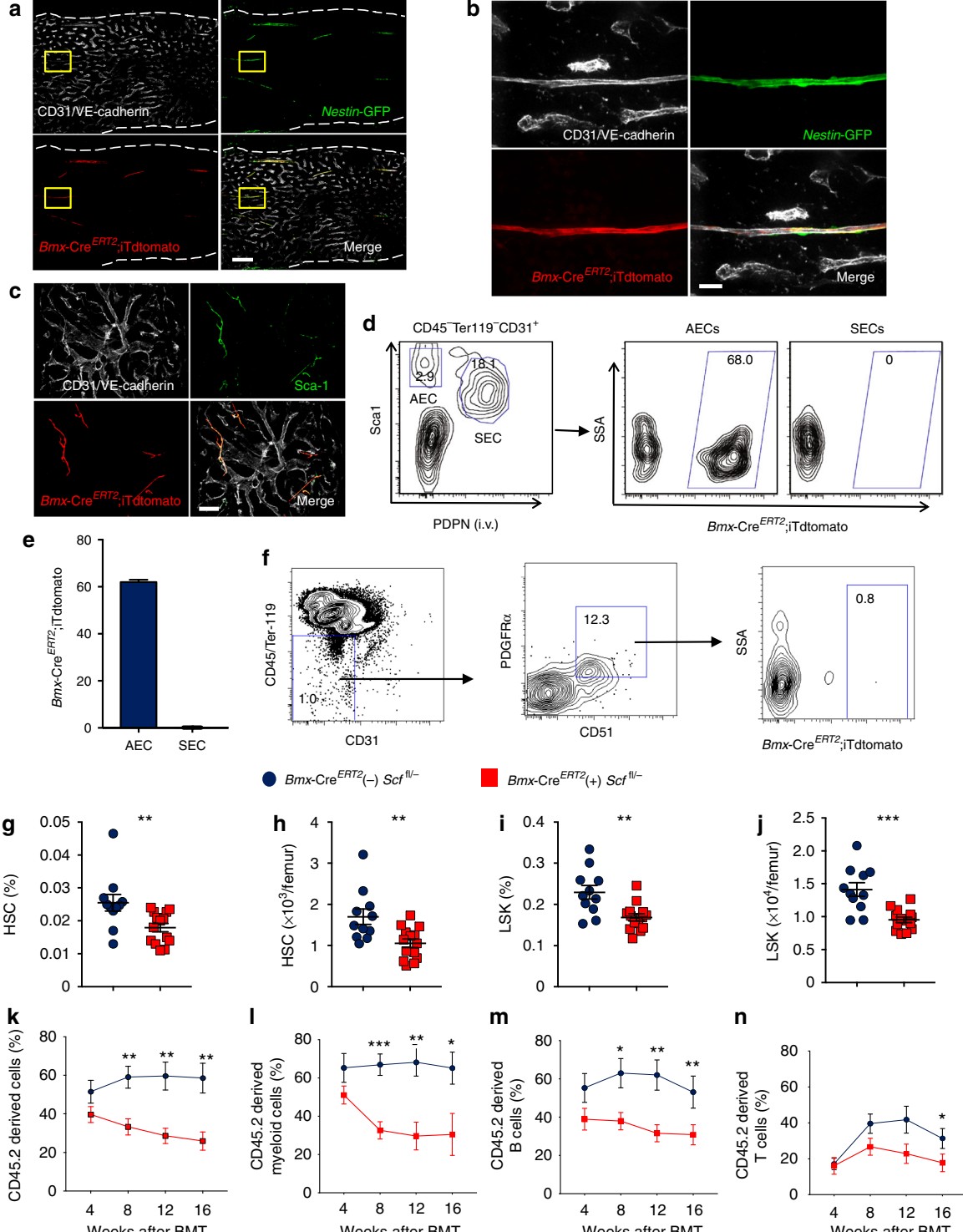

**Fig. 5** SCF from AECs regulates HSCs. **a** Representative images of femur from *Bmx*-Cre$^{ERT2}$;iTdtomato;*Nestin*-GFP mice stained with anti-VE-cadherin and anti-CD31. All the panels show the same field for different channels. Scale bar, 100 μm. **b** Enlarged image of the field highlighted by the yellow rectangle in (**a**). All panels show the same field for different channels. The *Bmx*-Cre$^{ERT2}$-labelled cells colocalise with *Nestin*-GFP-labelled cells. Scale bar, 10 μm. **c** Image of whole-mount sternal BM from *Bmx*-Cre$^{ERT2}$;iTdtomato injected with anti-Sca-1. Scale bar, 50 μm. **d** Representative FACS plot of the labelling of AECs and SECs by *Bmx*-Cre$^{ERT2}$. **e** Quantification of the labelling efficiency of AECs and SECs by *Bmx*-Cre$^{ERT2}$. Data are represented as mean ± SEM. $n = 8$ for AECs and SECs. **f** Representative FACS plot of *Nestin*-GFP$^+$ MSCs labelling by *Bmx*-Cre$^{ERT2}$. **g** Percentage of HSCs (CD150$^+$ CD48$^-$ LSK) in the BM. Data are represented as mean ± SEM. $n = 11$ for Cre (−) and $n = 14$ for Cre (+). **h** Number of HSCs (CD150$^+$ CD48$^-$ LSK) in the BM. Data are represented as mean ± SEM. $n = 11$ for Cre (−) and $n = 14$ for Cre (+). **i** Percentage of LSK in the BM. Data are represented as mean ± SEM. $n = 11$ for Cre (−) and $n = 14$ for Cre (+). **j** Number of LSK in the BM. Data are represented as mean ± SEM. $n = 11$ for Cre (−) and $n = 14$ for Cre (+). **k–n** Percentage of donor-derived cells after competitive transplantation. $n = 6$ mice for Cre (−), and $n = 7$ for Cre (+). The label of *X*-axis indicates the time (weeks) after transplantation. **g–n** Data were analysed with two-tailed *t*-test. *$p < 0.05$, **$p < 0.01$, ***$p < 0.001$, ****$p < 0.0001$

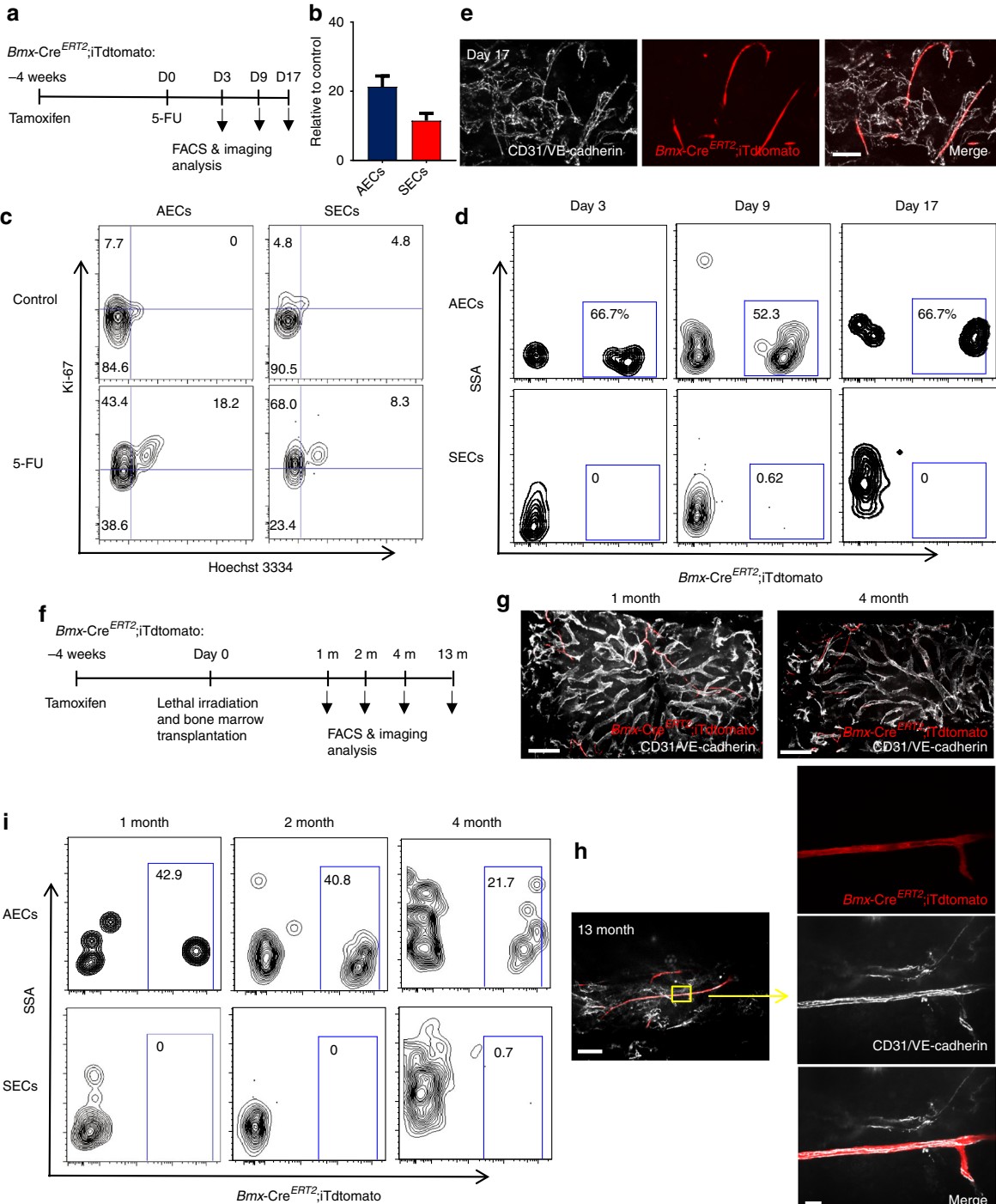

**Fig. 6** AECs self-regenerate, and do not regenerate SECs. **a** Scheme of experiment design. *Bmx*-Cre[ERT2];iTdtomato mice were injected with Tamoxifen to activate Cre expression. Four weeks later, 5-FU was given to these mice and BM ECs were analysed by FACS and immunofluorescence analysis. **b** Numbers of AECs and SECs 3 days after 5-FU treatment. Data are represented as mean ± SEM. **c** Representative FACS plot of cell cycle analysis of AECs and SECs from control mice and mice treated with 5-FU (9 days after 5-FU) using Hoechst 3334 and Ki-67. **d** FACS plot of the labelling of AECs and SECs by *Bmx*-Cre[ERT2] at different time points after 5-FU treatment. **e** Representative image of whole-mount sternum from mice treated as in (**a**). The bone was harvested on day 17 after 5-FU injection. Mice were injected i.v. with anti-VE-cadherin and anti-CD31. All panels show the same area for different channels. Scale bar, 50 μm. **f** Scheme of experiment design. *Bmx*-Cre[ERT2];iTdtomato mice were injected with Tamoxifen to activate Cre expression. Four weeks later, these mice were lethally irradiated and transplanted with BM cells from wild-type mice. BM ECs were analysed by FACS and immunofluorescence analysis at different time points after lethal irradiation. **g** Representative images from whole-mount sternum at 1 month (left panel) and 4 months (right panel) after lethal irradiation. Mice were injected i.v. with anti-VE-cadherin and anti-CD31. Scale bar, 200 μm. **h** Images from whole-mount sternum in which ECs are stained i.v. by anti-VE-cadherin and anti-CD31 injection at 13 months after lethal irradiation. The panels on the right show the same field for different channels. Scale bar, 100 μm in the left panel and 10 μm in the right panel. **i** FACS plot of the labelling of AECs and SECs by *Bmx*-Cre[ERT2] at different time points after lethal irradiation

remained exclusively in the AECs compartment during regeneration (Fig. 6d). Immunofluorescence analyses also revealed that iTdtomato[+] cells were confined to AECs 17 days after 5-FU challenge (Fig. 6e). These data suggests that AECs and SECs may regenerate separately after 5-FU-induced regeneration.

To study the vascular regeneration over a longer timeframe, we lethally irradiated tamoxifen-treated $Bmx$-Cre$^{ERT2}$; iTdtomato mice and transplanted them with wild-type BM cells for haematopoietic rescue (Fig. 6f). Animals were evaluated at various time points after a lethal dose of irradiation. Similar to the 5-FU model, we found that labelled ECs remained within the AECs compartment and did not contribute to SECs regeneration even 13 months post-irradiation (Fig. 6g, h). FACS analysis of BM ECs also showed that the $Bmx$-Cre$^{ERT2}$-labelled cells only gave rise to AECs (Fig. 6i). Taken together, these results indicate that AECs and SECs self-regenerate independently from specified radio-resistant endothelial precursors in the adult BM.

**AEC-derived SCF regulates HSC regeneration.** To investigate whether AECs-derived SCF contributes to HSC regeneration after myeloablation, we treated the $Bmx$-Cre$^{ERT2}$ (+) $Scf^{fl/-}$ mice and $Bmx$-Cre$^{ERT2}$ (−) $Scf^{fl/-}$ with 5-FU and analysed HSCs recovery at different time points after treatment (Fig. 7a). We found that HSC regeneration was significantly impaired in $Bmx$-Cre$^{ERT2}$ (+) $Scf^{fl/-}$ mice compared with $Bmx$-Cre$^{ERT2}$ (−) $Scf^{fl/-}$ at day 21 after 5-FU treatment (Fig. 7b). Furthermore, reconstitution capacity of day 21 HSCs from $Bmx$-Cre$^{ERT2}$ (+) $Scf^{fl/-}$ mice was significantly reduced compared to HSCs from $Bmx$-Cre$^{ERT2}$ (−) $Scf^{fl/-}$ (Fig. 7c). Thus, AEC-derived SCF contributes to HSC regeneration after 5-FU treatment.

## Discussion

Recent studies have suggested that multiple stromal and haematopoietic cells contribute to HSC maintenance. Although HSCs are broadly distributed close to sinusoids, the exact role of SECs in HSC maintenance has not been specifically demonstrated. Cultured AKT-activated ECs (which are likely a mixture of AECs and SECs) can expand HSPCs both in vitro and in vivo[29,30]. Specific deletion of the Notch1 ligand $Jag1$ in all ECs using constitutive $Cdh5$-Cre reduced BM HSC numbers in steady state and impaired haematopoietic recovery after sublethal irradiation[31]. Deletion of $Scf$ in the both the endothelial ($Tie2$-Cre) and perivascular stromal ($Lepr$-Cre) compartments have suggested additive contributions of these niches in that the reduction of HSC numbers in steady-state BM was greater when $Scf$ was deleted in both compartments compared to either alone[7]. Owing to the similar spatial distribution of CAR cells and SECs in the BM, it has been assumed that SECs might be the most important EC type for HSC maintenance. Our results show an unanticipated selective expression of niche factors in the arterial vascular tree, supporting an important role for an arterial-associated HSC niche.

The presence of an arterial niche for HSCs has been controversial. A subset of CD150[+] CD48[−] Lin[−] CD41[−] HSCs was shown to significantly associate to arterioles compared to the random assignment of HSC-like dots[11]. Importantly, the proportion of arteriole-associated HSCs was altered during their proliferation, suggesting a functional relevance[11]. Differences in permeability of sinusoids and arteries have subsequently been proposed to regulate HSC proliferation whereby less permeable arterial vessels maintained an environment low in reactive oxygen species (ROS) promoting HSC quiescence, while the more permeable sinusoidal vessels activated HSCs by high ROS due to plasma leakage[32]. Further studies show that megakaryocytes may

also represent a specific niche promoting HSC quiescence[33–35]. Others have argued that quiescent HSCs are uniformly distributed throughout the BM near sinusoids and that arteriole-associated stromal cells did not play a specific role in HSC maintenance[36]. The selective expression of SCF and other niche factors by AECs clearly support further the notion that arterioles confer a specific microenvironment for distinct HSC subsets. This idea is also consistent with recent studies showing that lineage-biased HSCs were located in distinct niches in which HSCs expressing von Willebrand factor (vWF) were associated with and regulated by megakaryocytes, whereas vWF[−] HSCs were found to be associated with and regulated by arteriolar perivascular cells[37]. It will be important in future investigations to address the implications of these distinct niches in the development and progression of haematological diseases.

## Methods

**Mice**. B6.Cg-Gt(ROSA)26Sortm14(CAG–tdTomato)Hze/J (iTdTomato) and Kitltm1.1Sjm/J ($Scf$-GFP) mice were purchased from Jackson Laboratory, $Cxcl12$-GFP mice and $Cxcl12^{fl/fl}$ mice were a gift from T. Nagasawa[23], and $Scf^{fl/fl}$ mice were a gift from S. Morrison[7]. C57BL/6J-Tg($Bmx$-Cre$^{ERT2}$)1Rha mice were kindly provided by Dr. Ralf Adams[28]. $Epor$-merCre mice were obtained from Dr. Ann Mullally[26]. $Nestin$-GFP mice are bred in our facilities[38]. C57BL/6-CD45.1/2 congenic strains were purchased from Jackson Laboratory. All mice were maintained in pathogen-free conditions under a 12 h:12 h light/dark cycle and fed ad libitum. All experiments were carried out using gender-matched littermate controls. All mice were analysed at 6–14 weeks of age. Both genders were used for experiments. All experimental procedures were approved by the Animal Care and Use Committees of Albert Einstein College of Medicine.

**Tamoxifen treatment**. For induction of $Bmx$-Cre$^{ERT2}$-mediated recombination, we adopted the methods used by the lab generating this mice to target arteries with modification[28]. Briefly, 2-week-old mice were injected with tamoxifen (Sigma) intraperitoneally at 100 mg kg$^{-1}$ dissolved in corn oil (Sigma) daily for two rounds of 5 consecutive days with a 7-day interval.

**Competitive repopulation assays**. Competitive transplantation was performed using the CD45.1/CD45.2 congenic system. Same number of BM cells ($3 \times 10^5$) collected from gene-deleted mice or control mice (CD45.2) were transplanted into lethally irradiated (6Gy + 6Gy) CD45.1 recipients with $3 \times 10^5$ competitor CD45.1 cells. Peripheral blood collection was carried out by retro-orbital plexus bleeding with a heparinised microcapillary tube under deep anaesthesia to 0.5 M EDTA solution. CD45.1/CD45.2 chimerism of recipients' blood was analysed for up to 4 months after transplantation.

**Flow cytometry and cell sorting**. The processing of BM cells were previously reported[10]. Briefly, for the analyses of haematopoietic cells, BM cells were flushed and dissociated by gently passing through a 21G needle. For analysis of stromal cells and ECs, BM cells were flushed and digested with 1 mg ml$^{-1}$ collagenase IV (Gibco), and 2 mg/ml dispase (Gibco) in Hank's balanced salt solution (Gibco) for 40 min at 37 °C. Ammonium chloride was used for red blood cell (RBC) lysis. Then cells were filtered through 100 μm nylon mesh. For FACS analysis, cells were stained with antibodies in PEB (PBS containing 0.5% BSA and 2 mM EDTA) buffer for 30 min at 4 °C. The following antibodies were used: allophycocyanin (APC)-anti-Gr-1 (eBioscience, Cat# 17-5931, clone RB6-8C5), phycoerythrin (PE)-anti-CD11b (eBioscience, Cat# 12-0112-83 clone M1/70), APC-eFluor780-anti-CD45R (eBioscience, Cat# 47-0452, clone RA3-6B2), PerCP-Cyanine5.5-anti-CD3e (eBioscience, Cat# 45-0031, clone 145-2C11), biotin-anti-Lineage (BD Biosciences, Cat# 559971, clones TER119, RB6-8C5, RA3-6B2, M1/70, 145-2C11), fluorescein isothiocyanate (FITC)-anti-CD45.2 (eBioscience, Cat# 11-0454, clone 104), FITC-anti-Ly6A/E (eBioscience, Cat# 11-5981, clone D7), PE-Cy7-anti-CD117 (eBioscience, Cat# 25-1171, clone 2B8), Brilliant Violet 421-anti-mouse CD48 (Biolegend, Cat# 103427, clone HM48-1), Brilliant Violet 605-anti-mouse CD150 (Biolegend, Cat# 115927, clone TC15-12F12.2), PE-Cy7-anti-Ki67 (eBioscience, Cat# 25-5698-80, clone SolA 15), PE-Cy7-anti-CD31 (eBioscience, Cat# 25-0311, clone 390), APC-eFluor780-anti-CD45 (eBioscience, Cat# 47-0451, clone 30-F11), APC-eFluor780-anti-TER119 (eBioscience, Cat# 47-5921, clone TER119), Alexa Fluor 700-anti-CD16/32 (eBioscience, Cat# 56-0161-82, clone 93), PerCP-Cyanine5.5-anti-CD127 (eBioscience, Cat# 45-1271-80, clone A7R34), PE-anti-CD135 (eBioscience, Cat# 12-1351, clone A2F10), streptavidin APC-eFluor780 (eBioscience, Cat# 47-4317-82), Alexa Fluor 647-anti-CD144 (Biolegend, Cat# 138006, clone BV13), Alexa Fluor 647-anti-CD31 (Biolegend, Cat# 102516, clone MEC13.3), PE-anti-mouse PDPN (Biolegend, Cat# 127408, clone 8.1.1), PE-Cy7-anti-mouse PDPN (Biolegend, Cat# 127412, clone 8.1.1), Alexa Fluor 488-anti-mouse CD31 (Biolegend, Cat# 102514, clone MEC13.3), Pacific Blue-anti-mouse CD54 (Biolegend, Cat# 116116, clone YN1/1.7.4), PE-Rat Anti-Mouse CD62E (BD,

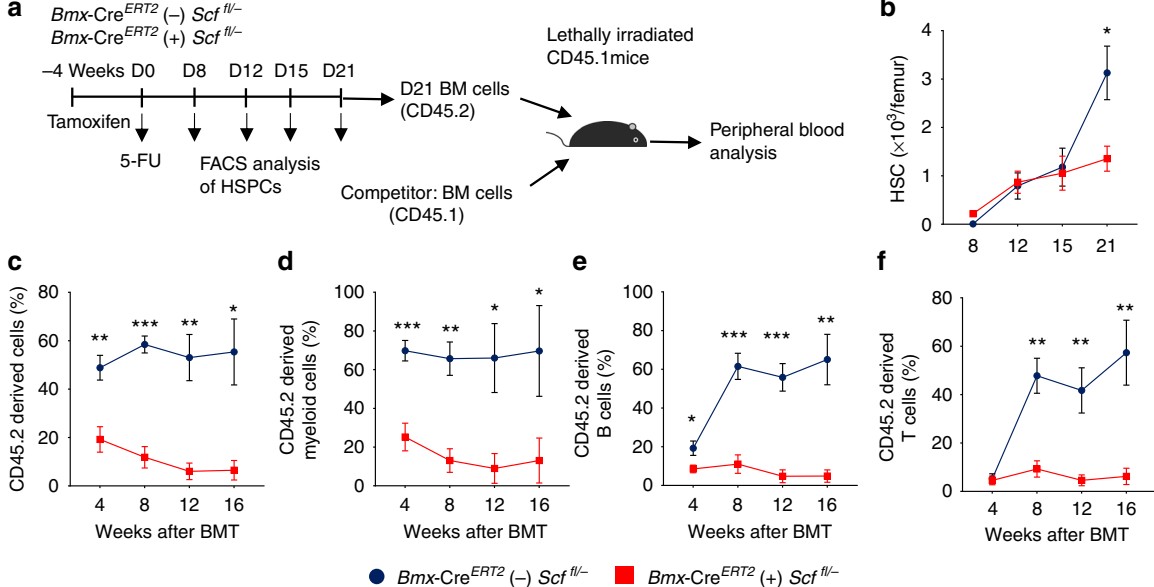

**Fig. 7** SCF from AECs contributes to HSCs regeneration. **a** Scheme of experiment design. *Bmx*-CreERT2;*Scf* ^fl/−^ mice and control mice were firstly given Tamoxifen to activate Cre expression. Four weeks later, 5-FU was administered and HSCs were analysed at different time points by FACS. **b** HSC number in femur at indicated time points. $n = 12$ mice for Cre (−), and $n = 9$ for Cre (+). **c**–**f** Percentage of donor-derived cells after competitive transplantation. $n = 5$ mice for Cre (−), and $n = 7$ for Cre (+). The label of X-axis indicates the time (weeks) after transplantation

Cat# 553751, clone 10E9.6), Alexa Fluor 700-Anti-mouse Ly-6A/E (Sca-1) (eBioscience, Cat# 56-5981-82, clone D7), PerCP-Cyanine5.5-anti-CD45 (eBioscience, Cat# 45-0451-82, clone 30-F11), PerCP-Cyanine5.5-anti-Ter119 (eBioscience, Cat# 45-5921-82, clone TER119), mouse EphB4 Antibody (R&D, Cat# AF446), Donkey anti-Goat IgG (H+L) Secondary Antibody, Alexa Fluor 488 (Thermo Fisher Scientific, Cat#A-11055), and PE Syrian Hamster IgG Isotype Ctrl Antibody (Biolegend, Cat# 402008, clone SHG-1). We used these antibodies at 1:100 dilution for FACS analysis. For analysis of EC by FACS, we injected i.v. 3 µg PE-anti-mouse PDPN or PE-Cy7-anti-mouse PDPN (Biolegend, Cat# 127412, clone 8.1.1) to the mice and waited for 10 min before we sacrificed the mice for FACS analysis. FACS analyses were carried out using an LSR II flow cytometer equipped with FACS Diva 6.1 software (BD Biosciences). Dead cells and debris were excluded by FSC, SSC, and 4′, 6-diamino-2-phenylindole (DAPI) (Sigma) staining. Cell sorting was performed on a FACSAria Cell Sorter (BD Biosciences). Data were analysed with FlowJo (Tree Star) software.

**RNA isolation and quantitative PCR.** Sorted cells were collected in Trizol (Sigma) and RNA was extracted following the manufacturer's protocol with one modification: linear acrylamide was added to the solution to facilitate RNA precipitation. Reverse transcriptions were performed using RNA to cDNA EcoDry Premix (Clontech) according to the manufacturer's protocols. Quantitative PCR was performed with SYBR Green (Roche) on the QuantStudio 6 Flex Real-time PCR System (Applied Biosystems). The PCR protocol consisted of one cycle at 95 °C (10 min) followed by 40 cycles of 95 °C (15 s) and 60 °C (1 min). Expression of *Gapdh* was used as an internal control. The relative gene expression is determined as $2^{-(Ct_{(gene)}-Ct_{(control)})}$. Primers used include: *Gapdh* forward, 5′-CAC ATT GGG GGT AGG AAC AC-3′; *Gapdh* reverse, 5′-ACC CAG AAG ACT GTG GAT GG-3′; *Cxcl12* forward, 5′-CGC CAA GG TCG TCG CCG-3′; *Cxcl12* reverse, 5′-TTG GCT CTG GCG ATG TGG C-3′; *Scf* forward, 5′-CCC TGA AGA CTC GGG CCT A-3′; *Scf* reverse, 5′-CAA TTA CAA GCG AAA TGA GAG CC-3′; *Bmx* forward, 5′-GCA AAA AGA GAT ACG GGG CAA-3′; *Bmx* reverse, 5′-GAA CTT TCC ATC CAC AAA GAA GC-3′; *Efnb2* forward, 5′-ATT ATT TGC CCC AAA GTG GAC TC-3′; *Efnb* reverse, 5′-GCA GCG GGG TAT TCT CCT TC-3′; *Epor* forward, 5′-GGT GAG TCA CGA AGT CAT GT-3′; *Epor* reverse, 5′-CGG CAC AAA ACT CGA TGT GTC-3′; *Icam1* forward, 5′-GGA CCA CGG AGC CAA TTT C-3′; *Icam1* reverse, 5′-CTC GGA GAC ATT AGA GAA CAA TGC-3′; *Vcam1* forward, 5′-GAC CTG TTC CAG CGA GGG TCT A-3′; *Vcam1* reverse, 5′-CTT CCA TCC TCA TAG CAA TTA AGG TG-3′; *Selp* forward, 5′-GGT ATC CGA AAG ATC AAC AAT AAG TG-3′; *Selp* reverse, 5′-GTT ACT CTT GAT GTA GAT CTC CAC ACA-3′; *Sele* forward, 5′-CCC TGC CCA CGG TAT CAG-3′; *Sele* reverse, 5′-CCC TTC ACA ACA GTC AAA CGT-3′.

**Cell cycle analyses.** Cell cycle analyses were performed using BD Cytofix/Cytoperm™ Fixation/Permeabilization Kit following the manufacturer's protocol. Briefly, BM cells were stained with surface markers, fixed and permeabilised in the

Fixation/Permeabilization solution for 20 min, washed, and stained with anti-Ki-67 antibody and Hoechst 33342 (Sigma) at 20 µg ml⁻¹ in 1× BD Perm/Wash™ buffer for 30 min.

**Whole-mount BM confocal immunofluorescence analyses of HSCs.** Endogenous HSC staining in sternal BM was performed as previously described[11]. Briefly, Alexa Fluor 647-anti-CD144 (Biolegend, cat# 138006, clone BV13, 1:100) and Alexa Fluor 647-anti-CD31 (Biolegend, Cat# 102516, clone MEC13.3) were i.v. injected into mice to stain the vasculature. Sternal BM fragments were removed 10 min after antibody injection, fixed with 4% PFA for 30 min, rinsed with PBS for three times, and then blocked/permeabilised in PBS containing 20% normal goat serum and 0.5% Triton X-100 at room temperature. Primary antibodies were incubated for 2 overnights at room temperature. The primary antibodies used were biotin-anti-lineage (TER119, RB6-8C5, RA3-6B2, M1/70, 145-2C11, BD Biosciences, cat. No. 559971, 1:100), biotin-anti-CD48 (eBioscience, cat# 13-0481, HM48-1, 1:100), biotin-anti-CD41 (eBioscience, cat# 13-0411, MWReg30, 1:3000), PE-anti-CD150 (Biolegend, cat# 115904, clone TC15-12F12.2, 1:100). After rinsing the tissue with PBS, the tissues were incubated with streptavidin eFluor 450 (eBioscience, cat# 48-4317, 1:100).

**RNA preparation and next-generation sequencing.** Total RNA from sorted *Nestin*-GFP^bright^ AECs, *Nestin*-GFP^dim^ AECs and SECs was extracted using the RNAeasy Plus Micro kit (Qiagen). The integrity and purity of total RNA were assessed using an Aligent 2100 Bioanalyzer (Agilent Technologies). Complementary DNA was generated using the SMART-Seq v4 Ultra Low Input RNA Kit for Sequencing (Clontech Laboratories) from total RNA. The Nextera XT DNA Sample Preparation Kit (Illumina) was used for the preparation of DNA libraries. The libraries were then submitted for Illumina HiSeq2500 sequencing (Illumina) according to standard operating procedures.

**RNA-Seq analysis.** RNA-Seq results generated from Illumina HiSeq 2500 were processed following a recently developed reproducible pipeline by Single Cell Genomics and Epigenomics Core facility at Albert Einstein College of Medicine. Briefly, single-end sequencing reads were aligned to the mouse genome using Spliced Transcripts Alignment to a Reference (STAR)[39]. We next used htseq-count to assign aligned reads to genes. Reads per kilobase of transcript per million (RPKM) was used as the expression quantification method[40]. The RPKM matrix was log2-transformed.

To identify genes that could be reliably used for analysis of each cell population, we first filtered out genes that had a counts per million < 0.1 in at least seven of the nine samples. 17,232 genes remained after filtering. Next, we normalised for RNA composition by calculating scaling factors for each sample using calcNormFactors in edgeR (version 3.12.0). The default method for computing scaling factors in calcNormFactors, trimmed mean of *M*-values (TMM) between each pair of

samples, was used. The original library size, before normalising, was calculated as the sum of the counts in each sample. The effective library size of each sample is the product of the original library size and the scaling factor for a given sample. Next, the estimate Disp function in edgeR was used to calculate the dispersions of the negative binomial model using the quantile-adjusted conditional maximum likelihood (qCML) method. Gene-wise exact tests were performed to test whether a gene was differentially expressed between any two conditions. Nominal $p$-values were corrected using the Benjamini–Hochberg procedure to adjust for multiple hypothesis testing.

**Statistics and reproducibility**. No statistical method was used to predetermine sample size. The experiments were not randomised and investigators were not blinded to allocation during experiments and outcome analyses. All data are represented as mean ± SEM. $n$ represents mouse number analysed in each experiment, as detailed in figure legends. We used two-tailed Student's $t$-tests for evaluating the significance of difference unless otherwise indicated. Statistical analyses were performed using GraphPad Prism 6 or 7 software. $*p < 0.05$, $**p < 0.01$, $***p < 0.001$, $****p < 0.0001$.

**Data availability**. RNA sequencing data have been deposited in the Gene Expression Omnibus (GSE104701). All data supporting the conclusions are available from the authors upon request.

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

## Acknowledgements

We thank C. Prophete and P. Ciero for technical assistance and L. Tesfa, Y. Wang, and D. Sun for help with cell sorting. We thank Drs. R. Adams, T. Nagasawa, and S. Morrison for providing genetically modified mice. This work was supported by R01 grants from the National Institutes of Health (NIH) (DK056638, HL069438, DK116312, DK112976 to P. S.F.). We are also grateful to the New York State Department of Health (NYSTEM Program) for shared facility (C029154) and research support (N13G-262) and the Leukemia and Lymphoma Society's Translational Research Program.

## Author contributions

C.X. designed and performed experiments, analysed and interpreted data, and wrote the manuscript. X.G., Q.W., and F.N. performed experiments. S.E.Z. and J.M. did the analysis of the RNAseq data. P.S.F. designed and supervised the study, interpreted data, and wrote the manuscript.

## Additional information

**Competing interests:** The authors declare no competing interests.

