## [Peer Review File · Nature Communications]

Reviewers' comments:

Reviewer #1 (Remarks to the Author):

Stem cell factor is secreted by arterial, but not sinusoidal, endothelial cells in the bone marrow

In this study Xu et al characterize BM AEC and SEC, adding new markers to this field. Segregating Ter119-/CD45-/CD31+ BM cells, their study defines AEC with the expression profile of Sca-1^{bright}/PDPN-/SCF^{high}/CXCL12+/Bmx+ while SEC are defined as Sca-1^{dim}/PDPN+/EpoR+/CXCL12-/SCF^{low} cells.

In the second part of their study. Xu et al deplete SCF expression by both types of BM blood vessels. Deletion of SCF expression specifically on AEC reduced repopulation potential of BM HSC, while no such effect was recorded for deletion of SEC-derived SCF which express only very low levels of this ligand.

In the third part of this study, Xu et al interrogated into BM vessel regeneration following 5-FU toxicity or irradiation-induced toxicity. They show that both blood vessel types proliferate following these treatments. They followed BmxCreERT2/iTdTomato and found this expression type in AEC only, suggesting that AECs and SECs regenerate independently.

Major comments:

1. The MS is not written in a clear and easy to follow way, rather it is complicated and confusing combining different topics which are not well connected together.

The first topic is an updated identification of bone marrow (BM) AECs versus SECs by several methods such as mRNA seq. profile, flow cytometry and imaging of BM sections. No functional role for the newly described markers is reported. The authors need to cite and to discuss what is already known about the BM vasculature and various blood vessels, markers which define them, their function etc. as detailed in the relevant comments below.

2. The second part contains a functional aspect which is the main finding of this study and also serves for the title, showing that SCF deletion specifically from AEC is harmful for BM HSC maintenance, while no such effect is exerted when SCF is depleted from SECs. However, SCF was already shown to be important for HSC function, and moreover, the very low levels of SCF expression by SECs to begin with, underscore the importance and merit of this selective depletion. Mechanistic investigation into the molecular mechanisms linking SCF depletion to this reduction in BM HSC could become a major part of the study. In addition, the authors should discuss their new findings over the published study by Ding et al (Nature 2012) which has already shown SCF deletion from BM endothelial cells effects on BM HSC, as SEC hardly express this cytokine.

3. The regenerative part of the study shows that following either 5-FU or irradiation treatments both AECs and SECs begin to proliferate. The authors conduct long-term tracking of Bmx-CreERT2;iTdTomato and show that all belong to AECs. The outcome in the SEC compartment looking at markers of these cells is not shown. Maybe the text should better explain the authors' conclusion as to the independent regeneration of both EC types. Nevertheless, the biological meaning of this observation is not studied and the results are only preliminary.

4. The authors suggest that currently accepted FACS method of AEC vs SEC definition based on Sca-1, CD31, CD45, TER119 markers is inaccurate and describe better ways to define AEC vs SEC. Their conclusion regarding Sca-1-CD31+(CD45-TER119-) population, obtained by FACS of in vitro stained digested BM, being actually a non-endothelial population is based on the fact that when they inject anti CD31 ab in vivo they see that all the vessels are Sca1+ by FACS. The authors suggest podoplanin/ICAM-1/Selectin as better markers of sinusoids instead or in addition to Sca-1. I think that this methodological conclusion is important for the field of BM vessel characterization. But the authors should use additional accepted sinusoidal markers in addition such as Endoglin, VEGFR3, EPHB4 receptor etc. I would also suggest the authors to consider as well the definition by FACS of transition vessels. This for instance could be done by addition of Endomucin ab. Would these transition vessels have Podoplanin/ICAM/E selectin? And what is the level of Sca-1 in these Type H vessels? Please revisit and take a closer look on Morrison and Ralf Adams MS on these

topics and discuss them accordingly.

5. The authors claim that CXCL12 is expressed more on BM AECs than on SECs. This data only confirms previously published results by Itkin et al (Nature, 2016) which has already shown that AEC express higher levels of CXCL12 compared to SEC. Please revisit citation and cite accordingly.

6. The authors should add additional specific markers for sinusoidal ECs. VE-cad labels all the endothelial cells, and is not specific for sinusoids. For example: the group of Tosato G. et al (JCI, 2016) show that EPHB4 receptor is functionally expressed by BM sinusoids BM; SECs are Endomucin+/EPHB4+/Sca-1-/low, while AECs are Endomucin-/EPHB4-/low/Sca-1+. Please cite and discuss this paper.

7. The authors noted that the erythropoietin receptor (Epor) was selectively expressed on SECs (Fig. 2g). They claim that Epor expression on SECs has not been previously reported. The group of Minato et al (cell structure and function, 2017) recently showed that bone marrow endothelial cells functionally express Epor. We suggest for the authors to cite and to discuss this paper.

8. The authors further characterize AECs vs SECs based on Nestin-GFP expression and find Bmx exclusively in AECs. Later on the authors develop Stem cell factor KO mouse in AECs based on expression of Bmx in these cells. However, it is not clear to me if these KO mice lack stem cell factor in transition vessels as well in addition to arterioles/arteries. The image of Bmx TdTomato show selected regions but it is not clear how the endosteal regions of diaphysis and metaphysis look like. The authors are encouraged to show informative stainings before and after depletion.

9. The authors claim that when the Scf is depleted in AECs it reduces the percentage of HSCs and also reduces their repopulation potential. This is an important observation that could be further strengthened by showing that HSCs in these mice are also less chemotherapy resistant and/or less irradiation resistant. The authors imply that AECs could be part of the stem cell niche. However, it would be interesting to see images of HSCs residing near the AECs. Moreover, it would be interesting to see if in Scf KO mouse (KO in AEC) the localization of these HSCs changes or if it is reduced.

Minor comments:

1. The authors claim that AECs can't be separated by FACS because SEC also express Sca-1. However, based on their own previous published results (Kunisaki Nature 2013), SEC are Sca-1low while arterioles are Sca-1high.

2. The authors show that SECs expressed higher levels of Stab2, Dnase1l3, and C1qtnf1, as well as adhesion molecules mediating haematopoietic stem and progenitor cell homing, Sele, Selp, Icam1, Vcam1 than aECs (Fig. 2g). The higher expression of Sele, Selp, Icam1, Vcam1 in AECs (it should be SECs) compared to SECs (it should be AECs) was also confirmed using qPCR analysis (Supplementary Fig. 2b).

3. For the RNA seq, the authors show that Sca1 is expressed both in AEC and SEC. What about PDPN? The authors should show the expression of PDPN on AEC and SEC.

4. They claim that E-sel and ICAM can substitute PDPN. The authors should show the staining on bones sections, not only by FACS.

5. The authors claim that EpoR label 80% of the SAC. They should show the staining of PDPN and Epor Cre mice not only by FACS, but also by IHC. The same for the Bmx Cre mice, we should not be able to see the co-localization of PDPN with Bmx.

6. Not clear how AEC and SEC were gated in Fig 2A for the purpose of sorting for RNAseq analysis. If SECs and some AECs are both Nestin-GFP dim then how were they defined by FACS?

7. In bone images, the authors should specify in the legend if metaphysis or diaphysis are presented.

8. Line 189: please add reference

Reviewer #2 (Remarks to the Author):

This is an interesting study addressing the issue about the heterogeneity of endothelial cells (ECs) in bone and their contribution to the hematopoietic stem cell (HSC) niche. There is a bit of

controversy in the field regarding this question. While several researchers favor sinusoidal (capillary) endothelial cells as a main component of the HSC niche, other studies, such as the current manuscript, argue for arterial ECs and/or other EC subpopulations.

In the current manuscript, Xu et al. have separated the arterial (AECs) and sinusoidal ECs (SECs) by FACS with a combination of antibodies. AECs and SECs show distinct gene expression signatures, and contribute differentially to HSC maintenance and BM regeneration. Overall, this is an interesting and timely study, but there are several important issues that need to be addressed prior to publication of the data.

Questions:

1) According the RNA-seq data, *Kitl* is expressed in SECs even though the level is lower than in AECs. This obviously is not consistent with the message in the title of the study. At present, the data only support the conclusion that SCF from AECs is required for HSC niche function. Along the same lines, the authors provide evidence that AECs have more SCF than the SEC bulk population at the transcript level (Fig. 2h and 2i). However, it is quite possible that SECs are heterogenous so there could be a subset of cells with higher SCF expression. SCF was actually detectable in SECs based on RNA-seq in Fig. 2h. The authors need to show absolute RPKM/FPKM value of *Kitl* (and also *Cxcl12*) so that readers can form their own opinion on this important issue.

2) I have concerns about the purity of the isolated ECs. If *Epor-Cre* only labels SECs and *Bmx-CreER* only AECs (based on imaging), the FACS data indicate that about 20% to 35% cells do not correspond to real arterial ECs (Fig. 3c and 4d). Moreover, does *Epor-Cre* label other capillary EC subpopulations in the metaphysis, which is not shown in the current data set. What about *Epor-Cre* recombination in other cell populations? Better analysis of the genetic tools and the AEC/SEC populations is necessary.

Heinrich et al. (ref. 20) have reported *Epor-Cre* recombination predominantly in the Ter119-positive fraction, whereas less recombination was seen in Ter119-negative BM and spleen cells. This seems inconsistent with the current study.

3) *Epor-Cre* shows recombination in about 20% of AECs and therefore it is important to show that arterial SCF is not altered in the *EporCre*-induced SCF mutants. Conversely, the authors should show that SCF expression in SECs is not altered in the *Bmx-CreER*-generated mutants. Along the same lines, SCF-GFP and SCF (*Kitl*) transcripts are prominently expressed in perivascular *Lepr+* or CAR cells (Lei Ding et al., Nature 2012; Noboru Asada et al., NCB 2017). Is perivascular SCF expression altered in the EC-specific mutant models?

4) The authors argue that SCF from sinusoidal ECs is dispensable for HSC maintenance under physiological conditions. Does AEC or SEC-derived SCF control BM regeneration after irradiation or chemotherapy? The authors should use *Epor-Cre* and *Bmx-CreER*-specific SCF mutant mice to address this important question.

5) The FACS plots on the top and bottom right in Fig.1e are identical or almost identical (see dot distribution) even though different antibodies were used. The authors need to check the original data and correct this figure.

In Supplementary Fig. 2c and 2d, the overlay plots are not clearly described in the legends. What are the gating areas and how to red/blue dots correspond to each other? Please provide information on how the overlay picture was generated.

6) Several studies, including some by the lead author of this manuscript, have shown that arterioles maintain quiescent HSCs. Is this population specifically altered in the *Bmx-CreER*-generated SCF mutant?

Likewise, HSC numbers are changed in *Bmx-CreERT SCFfl/-* mutants but, strangely, this has no effect on cellularity and differentiation (Suppl. Fig.4). Please explain.

7) By combining Sca-1 and PDPN with CD45/Ter119/CD31, the authors separated AECs and SECs, which is fundamental for this work. In Fig. 1f, PDPN (i.v.) is indicated. But in the main text part related to this figure (line 89-95), PDPN is done by ex vivo staining, not intravenous labeling. Please make sure the correct experimental procedure is described and explain whether there could be differences between in ex vivo staining and intravenous labeling. The authors also need to provide more detail in the Methods (e.g. dosage, antibody catalog numbers, treatment duration).

8) In Fig. 2a, both AECs and SECs are Nestin-GFPdim. However, it is difficult to see GFPdim signal by immunostaining. Most of sinusoidal ECs appear to be GFP negative (Fig. 1d, 3a, 4a). Do Nestin-GFPdim (GFP+) ECs only represent a specific population of sinusoidal ECs? Please show Nestin-GFPdim staining in SECs in Fig. 1d, Fig. 3a and Fig. 4a.

Nestin-GFP also labels perivascular cells that are closely associated with ECs. The authors need to provide images with higher magnification (40x or 63x) to show co-localization in Fig. 1d, 3b and 4b.

Previous study from the authors showed Nestin-GFP positive cells are CD31-negative and CD34, VE-cadherin-negative (Mendez-Ferrer et. al., Nature 2010). The current study shows that Nestin-GFP cells are positive for CD31 and VE-Cad. Please explain.

Minor points:

9) Something is wrong in the text referring to Supplemental Fig. 2b. Please check and correct.

10) In Fig. 2k, please explain the meaning of the label "PE" on the vertical axis. In Fig. 2l, the label for the vertical axis and the percentage of gated cells are missing.

11) In Fig. 4d, please show quantitative data for the percentage of AECs in Bmx-CreER+ cells.

12) In Fig. 5i, after irradiation, the percentage of Bmx-CreER+ cells in AECs decreased at 4 months (21.7%) relative to 1 month (42.9%). But in Fig. 5g, Bmx-CreER+ cells seem increased at 4 months. Please explain.

In Fig. 5i, percentage of AECs/SECs is not indicated in the plots.

Reviewer #3 (Remarks to the Author):

Overall the manuscript is well written and the conclusions supported by the data. The manuscript is of great interest to the field and will substantially increase the understanding of the different types of vasculature within the BM.

However, some of the language of the manuscript is very definite, yet this is not proven by the data and should be toned down. For example, the authors commonly say AEC or SEC specific genes, yet all the data represents relative expression in one population to another, not that the gene is only expressed in this population and is not expressed in any other cell type in the body. As the latter is highly unlikely and has been proven to be untrue for many previously reported "specific genes", less definitive terminology would be appropriate.

In addition, no isotypes or other controls are included for any of the FACS analysis or immunohistochemistry. These should be included.

Importantly, the authors describe the populations as CD45-Ter119, yet in the vast majority of the dotplots in the manuscript showing CD45Ter119 versus CD31, a real proportion (or even most) of the population labelled as CD45Ter119- would have to be considered CD45Ter119dim. An isotype and an unlabeled control would be very helpful here, but the authors should discuss this.

Minor points:

Line 119 and 120. The authors contradict themselves. In line 119 they state sele, selp etc. are more highly expressed in SEC than AEC, then in line 120, they reverse this and say higher in AEC than SEC. The data in sup 2b supports the statement in line 119.

We are very thankful to the reviewers for the very helpful and constructive suggestions. We have considered these comments very seriously and addressed all the points in the response below. The resulting changes in the manuscript have been highlighted blue ink to facilitate the review.

Reviewers' comments:

Reviewer #1 (Remarks to the Author):

Stem cell factor is secreted by arterial, but not sinusoidal, endothelial cells in the bone marrow

In this study Xu et al characterize BM AEC and SEC, adding new markers to this field. Segregating Ter119-/CD45-/CD31+ BM cells, their study defines AEC with the expression profile of Sca-1bright/PDPN-/ SCFhigh/CXCL12+/Bmx+ while SEC are defined as Sca-1dim/PDPN+/EpoR+/CXCL12-/SCFlow cells.

In the second part of their study. Xu et al deplete SCF expression by both types of BM blood vessels. Deletion of SCF expression specifically on AEC reduced repopulation potential of BM HSC, while no such effect was recorded for deletion of SEC-derived SCF which express only very low levels of this ligand.

In the third part of this study, Xu et al interrogated into BM vessel regeneration following 5-FU toxicity or irradiation-induced toxicity. They show that both blood vessel types proliferate following these treatments. They followed BmxCreERT2/iTdTomato and found this expression type in AEC only, suggesting that AECs and SECs regenerate independently.

Major comments:

1. The MS is not written in a clear and easy to follow way, rather it is complicated and confusing combining different topics which are not well connected together.

We thank the reviewer for the comment. We feel that the three parts that the reviewer has correctly identified are linked in that i) we have developed a novel approach to separate and isolate with high purity AECs and SECs in the bone marrow; ii) we have used this knowledge to show that niche factor expression (particularly SCF) is segregated in the AEC compartment and evaluated this issue functionally with genetic models; iii) we have then used the AEC-specific genetic model to evaluate the role of AEC in the regeneration of the bone marrow vasculature. We have edited the paper to improve the flow and have added subheadings to guide the reader through the text.

The first topic is an updated identification of bone marrow (BM) AECs versus SECs by several methods such as mRNA seq. profile, flow cytometry and imaging of BM sections. No functional role for the newly described markers is reported. The authors need to cite

and to discuss what is already known about the BM vasculature and various blood vessels, markers which define them, their function etc. as detailed in the relevant comments below.

We thank the reviewer for the suggestion to add more information about Podoplanin (PDPN)'s function. PDPN is mucin-type transmembrane glycoprotein expressed on lymphatic endothelial cells and fibroblastic reticular cells. Deletion of PDPN in mice disrupted normal lymphatic vasculature formation and caused death immediately after birth due to respiratory failure (Schacht V et al. *EMBO J.* 2005). PDPN has been suggested maintains high-endothelial venule integrity by interacting with platelet CLEC-2 (Herzog et al *Nature* 2013). We have revised the manuscript to include the background information. It will be interesting in future studies to investigate the role of PDPN in sinusoids in the bone marrow which are the site of hematopoietic cell migration, as sinusoid PDPN may regulate the HSC and leukocytes trafficking.

2. The second part contains a functional aspect which is the main finding of this study and also serves for the title, showing that SCF deletion specifically from AEC is harmful for BM HSC maintenance, while no such effect is exerted when SCF is depleted from SECs. However, SCF was already shown to be important for HSC function, and moreover, the very low levels of SCF expression by SECs to begin with, underscore the importance and merit of this selective depletion. Mechanistic investigation into the molecular mechanisms linking SCF depletion to this reduction in BM HSC could become a major part of the study. In addition, the authors should discuss their new findings over the published study by Ding et al (*Nature* 2012) which has already shown SCF deletion from BM endothelial cells effects on BM HSC, as SEC hardly express this cytokine.

We have discussed in more detail the study of Ding et al (*Nature* 2012) in the revised manuscript. In their study, constitutive *Tie2-Cre* mice were used to delete *Scf* in both AECs and SECs starting from the developmental stage. They have also found that the contribution in SCF synthesis and impact on HSC numbers in bone marrow was similar between ECs and deletion in stromal cells using *Lepr-Cre*. Deletion of *Scf* using both *Tie2-Cre* and *Lepr-Cre* led to a more severe phenotype. Here, we show that SCF expression is not homogeneous in ECs and that, surprisingly, SECs express very little SCF compared to AECs. These results are supported with specific genetic deletion experiments. These data add significantly to the literature because it has been thought previously that SECs provided most EC-derived HSC niche activity in the bone marrow. The results are thus novel, important, and unexpected.

3. The regenerative part of the study shows that following either 5-FU or irradiation treatments both AECs and SECs begin to proliferate. The authors conduct long-term tracking of *Bmx-CreERT2*; *iTdTomato* and show that all belong to AECs. The outcome in the SEC compartment looking at markers of these cells is not shown. Maybe the text should better explain the authors' conclusion as to the independent regeneration of both EC types. Nevertheless, the biological meaning of this observation is not studied and the results are only preliminary.

We have edited the text to reflect the reviewer's suggestion. We have also explained the conclusion more clearly with the subheading "AECs self-regenerate and do not regenerate the sinusoid compartment". In addition, we have carried out additional experiments on the contributions of SCF derived from AECs and SECs in the regeneration of HSCs after myeloablation with 5-FU. We have found that SCF produced by AECs promoted HSC regeneration. The new data were added to Fig. 6.

4. The authors suggest that currently accepted FACS method of AEC vs SEC definition based on Sca-1, CD31, CD45, TER119 markers is inaccurate and describe better ways to define AEC vs SEC. Their conclusion regarding Sca-1-CD31+(CD45-TER119-) population, obtained by FACS of in vitro stained digested BM, being actually a non-endothelial population is based on the fact that when they inject anti CD31 ab in vivo they see that all the vessels are Sca1+ by FACS. The authors suggest podoplanin/ICAM-1/Selectin as better markers of sinusoids instead or in addition to Sca-1. I think that this methodological conclusion is important for the field of BM vessel characterization. But the authors should use additional accepted sinusoidal markers in addition such as Endoglin, VEGFR3, EPHB4 receptor etc. I would also suggest the authors to consider as well the definition by FACS of transition vessels. This for instance could be done by addition of Endomucin ab. Would these transition vessels have Podoplanin/ICAM/E selectin? And what is the level of Sca-1 in these Type H vessels? Please revisit and take a closer look on Morrison and Ralf Adams MS on these topics and discuss them accordingly.

We thank the reviewer for the appreciation of the importance of our manuscript to the field. The reviewer brought to our attention a recent study (Kwak et al. J Clin Invest. 2016) who showed that EPHB4 could mark bone marrow sinusoids by immunofluorescence analysis. To address the comment, we have stained enzymatically digested bone marrow with antibodies against Sca-1, PDPN, or EPHB4. We have found that PDPN and EPHB4 staining indeed overlap and can mark sinusoids. We have cited the reference, and added the results in Supplementary Fig. 1d. Although CD105, endomucin, and VEGFR3 are expressed at slightly higher levels on SECs than AECs, they cannot be used to separate AECs from SECs by FACS as shown in Reviewer Fig. 1 below.

For Reviewer Fig. 1:

For Reviewer Fig. 1

Enzymatically digested bone marrow cells were stained with anti-CD45, anti-Ter119, anti-CD31, anti-Sca-1, anti-PDPN (*i.v.*), and either anti-endomucin, anti-CD105 or anti anti-VEGFR3, and analysed by FACS.

With regards to FACS analysis of transition vessels, we have not found any reliable markers that can separate them from other endothelial cells by FACS. Endomucin is broadly expressed on sinusoids, AECs and type H ECs (Kusumbe et al. Nature 2014 and Acar et al. Nature 2015).

5. The authors claim that CXCL12 is expressed more on BM AECs than on SECs. This data only confirms previously published results by Itkin et al (Nature, 2016) which has already showed that AEC express higher levels of CXCL12 compared to SEC. Please revisit citation and cite accordingly.

We thank the reviewer for this comment. We had indeed carefully reviewed the Itkin et al paper. However, we were concerned about the separation of SECs from AECs by FACS in this study because: (1) SECs were considered to be Sca-1 negative, which is not the case as our results clearly show that all ECs express Sca-1. (2) the paper showed that AECs expressed *higher* levels of adhesion molecules such as VCAM-1, ICAM-1 and P-selectin than SECs (please see Extended Data Figure 3). As shown in Fig. 1 of our manuscript, Sca-1 expression is not sufficient to separate accurately AECs from SECs by FACS. After separation with PDPN, we have demonstrated using multiple methods (RNAseq, qPCR, FACS analysis) that SECs clearly express higher of adhesion molecules expression than AECs. In addition, we have added new immunofluorescence analysis of ICAM-1, E-selectin and P-selectin expression which confirms that the SECs express much higher adhesion molecule expression than AECs. Taking these results into consideration, the separation of AECs and SECs in the Itkin's paper may not have been accurate, which makes their analysis of CXCL12 expression in these subsets is difficult to interpret. For these reasons, we chose to discuss the other novels aspects of this paper in our manuscript.

6. The authors should add additional specific markers for sinusoidal ECs. VE-cad labels all the endothelial cells, and is not specific for sinusoids. For example: the group of Tosato G. et al (JCI, 2016) show that EPHB4 receptor is functionally expressed by BM sinusoids BM; SECs are Endomucin+/EPHB4+/Sca-1-/low, while AECs are Endomucin-/EPHB4-/low/Sca-1+. Please cite and discuss this paper.

Please see response to #4 comment above.

7. The authors noted that the erythropoietin receptor (Epor) was selectively expressed on SECs (Fig. 2g). They claim that Epor expression on SECs has not been previously reported. The group of Minato et al (cell structure and function, 2017) recently showed that bone marrow endothelial cells functionally express Epor. We suggest for the authors to cite and to discuss this paper.

We thank the reviewer for bringing this paper to our attention. We could not have cited the work because it came out 8 days after our original submission (17th October 2017). In this study, the authors show that the total bone marrow endothelial cells express Epor, but whether AECs or SECs, or both, express Epor was not investigated. Our

studies show that *Epor* is selectively expressed on SECs. We have cited this paper and revised the text accordingly.

8. The authors further characterize AECs vs SECs based on Nestin-GFP expression and find *Bmx* exclusively in AECs. Later on the authors develop Stem cell factor KO mouse in AECs based on expression of *Bmx* in these cells. However, it is not clear to me if these KO mice lack stem cell factor in transition vessels as well in addition to arterioles/arteries. The image of *Bmx* TdTomato show selected regions but it is not clear how the endosteal regions of diaphysis and metaphysis look like. The authors are encouraged to show informative stainings before and after depletion.

As mentioned in our response to comment #4, we were unable to observe transitional vessels due to lack of definitive markers in adult bone marrow. For the imaging of *Bmx-Cre^{ERT2};iTdTomato* mice, we have revised Fig. 4a to include the endosteal region of diaphysis and included the imaging of metaphysis in Supplementary Fig. 8. We have also updated the manuscript by showing the images of bone marrow from *Bmx-Cre^{ERT2};Scf^{fllox/-}* mice and control mice in Supplementary Fig. 9a. We have analyzed the AECs and SECs frequency in these mice (shown in Supplementary Fig. 9b). We did not see a significant difference.

9. The authors claim that when the *Scf* is depleted in AECs it reduces the percentage of HSCs and also reduces their repopulation potential. This is an important observation that could be further strengthened by showing that HSCs in these mice are also less chemotherapy resistant and/or less irradiation resistant. The authors imply that AECs could be part of the stem cell niche. However, it would be interesting to see images of HSCs residing near the AECs. Moreover, it would be interesting to see if in *Scf* KO mouse (KO in AEC) the localization of these HSCs changes or if it is reduced.

We thank the reviewer for the suggestion. We have carried out additional experiments with 5-FU-induced regeneration in the *Bmx-Cre^{ERT2};Scf^{fl/-}* and control mice. The results of these experiments have revealed that deletion of *Scf* in AECs delayed HSC regeneration and reduced the repopulation capacity after competitive bone marrow transplantation. The new data are shown in Fig. 6.

Our previous studies have shown that a subset of HSCs localizes near (or adjacent) to Nestin-GFP^{bright} arterioles (Y Kunisaki et al. *Nature* 2013). As shown in Fig. 4a, *Bmx-Cre^{ERT2}* targeted cells co-localize with Nestin-GFP^{bright} arteries/arterioles, and these HSCs are also close to *Bmx-Cre^{ERT2}* targeted AECs. As suggested by the reviewer, we have also performed HSC localization analysis in the control mice and *Bmx-Cre^{ERT2}; Scf^{fl/-}* mice and found that there was no difference in the localization of HSCs with or without *Scf* deletion in the AECs. The new results are shown in Supplementary Fig. 9g. In addition, we have performed HSC cell cycle analysis in the control and *Bmx-Cre^{ERT2}; Scf^{fl/-}* mice and did not see significant change in the percentage of cells in G0 phase. The new data are presented as Supplementary Fig. 9f.

Minor comments:

1. The authors claim that AECs can't be separated by FACS because SEC also express Sca-1. However, based on their own previous published results (Kunisaki *Nature* 2013), SEC are Sca-1^{low} while arterioles are Sca-1^{high}.

The reviewer is correct that Sca-1 expression can discern arterial from sinusoid vessels by immunofluorescence microscopy owing to the high level of expression and morphology of the vessels (Kunisaki et al. *Nature* 2013, Nombela-Arrieta et al *Nature Cell Biology* 2013). Although previous studies in the field have defined AECs as Sca-1⁺, and SECs as Sca-1⁻ (Itkin et al *Nature*, 2016; Hooper et.al *Cell Stem Cell*, 2010), our results clearly show that all ECs express Sca-1 by FACS and that this marker alone cannot reliably discriminate AECs from SECs without the addition of another marker. In general, AECs do express higher levels of Sca-1 than SECs, but our results show that a second marker is needed to separate them and prospectively isolate them with high purity.

2. The authors show that SECs expressed higher levels of Stab2, Dnase1l3, and C1qtnf1, as well as adhesion molecules mediating haematopoietic stem and progenitor cell homing, Sele, Selp, Icam1, Vcam1 than aECs (Fig. 2g). The higher expression of Sele, Selp, Icam1, Vcam1 in AECs (it should be SECs) compared to SECs (it should be AECs) was also confirmed using qPCR analysis (Supplementary Fig. 2b).

We thank the Reviewer for noticing the error which was corrected.

3. For the RNA seq, the authors show that Sca1 is expressed both in AEC and SEC. What about PDPN? The authors should show the expression of PDPN on AEC and SEC.

PDPN is selectively expressed on SECs by FACS and is not found on AECs (Fig. 1e). Similar results are observed by immunofluorescence analyses of whole-mount bone marrow (Fig. 1d). However, our RNAseq analysis shows that PDPN mRNA does not correlate with the protein expression (RPKMs of PDPN in SEC, Nestin^{dim} AECs, and Nestin^{bright} AECs are 0.08 ± 0.11 , 0.03 ± 0.05 , and 0.29 ± 0.50 , respectively). PDPN expression was reported to be regulated post-transcriptionally (Staines KA et al. *J Cell Physiol.* 2016; Eisenreich et al *Journal of Hypertension*, 2017), which may account for the differential protein expression of PDPN between AECs and SECs.

4. They claim that E-sel and ICAM can substitute PDPN. The authors should show the staining on bones sections, not only by FACS.

We thank the reviewer for the suggestion. We have carried out further analyses of adhesion molecule expression in whole-mount sternal immunofluorescence microscopy and found that indeed adhesion molecules P-selectin, E-selectin and ICAM-1 labels sinusoids specifically. The results were added as Supplementary Fig. 3d - f.

5. The authors claim that EpoR label 80% of the SEC. They should show the staining of PDPN and Epor Cre mice not only by FACS, but also by IHC. The same for the Bmx Cre mice, we should not be able to see the co-localization of PDPN with Bmx.

We have shown by immunofluorescence in Fig. 1d that PDPN is selectively expressed on sinusoid vessels but not on AECs. In Fig. 4a, b and c, we have used two imaging methods to identify arteries (Nestin-GFP and anti-Sca-1 *in vivo* staining), and show that *Bmx-Cre*^{ERT2} targets specifically arterial vessels. In Fig. 3a, we show that *Epor-Cre* targets sinusoids with very few segments of Nestin-GFP^{bright} arteries.

6. Not clear how AEC and SEC were gated in Fig 2A for the purpose of sorting for RNAseq analysis. If SECs and some AECs are both Nestin-GFP dim then how were they defined by FACS?

We apologize for the confusion. We have separated bone marrow endothelial cells into AECs and SECs using PDPN/Sca-1 expression. Then, we have further separated AECs into Nestin-GFP^{dim} and Nestin-GFP^{bright} fractions. SECs were not further separated because of their homogenous low Nestin-GFP levels.

7. In bone images, the authors should specify in the legend if metaphysis or diaphysis are presented.

We added this information to the revised manuscript.

8. Line 189: please add reference

Citation was added in the revised manuscript.

Reviewer #2 (Remarks to the Author):

This is an interesting study addressing the issue about the heterogeneity of endothelial cells (ECs) in bone and their contribution to the hematopoietic stem cell (HSC) niche. There is a bit of controversy in the field regarding this question. While several researchers favor sinusoidal (capillary) endothelial cells as a main component of the HSC niche, other studies, such as the current manuscript, argue for arterial ECs and/or other EC subpopulations.

In the current manuscript, Xu et al. have separated the arterial (AECs) and sinusoidal ECs (SECs) by FACS with a combination of antibodies. AECs and SECs show distinct gene expression signatures, and contribute differentially to HSC maintenance and BM regeneration. Overall, this is an interesting and timely study, but there are several important issues that need to be addressed prior to publication of the data.

We thank the reviewer for the appreciation of the importance and general interest of our manuscript.

Questions:

1) According the RNA-seq data, *Kitl* is expressed in SECs even though the level is lower than in AECs. This obviously is not consistent with the message in the title of the study. At present, the data only support the conclusion that SCF from AECs is required for HSC niche function.

Along the same lines, the authors provide evidence that AECs have more SCF than the SEC bulk population at the transcript level (Fig. 2h and 2i). However, it is quite possible that SECs are heterogenous so there could be a subset of cells with higher SCF expression. SCF was actually detectable in SECs based on RNA-seq in Fig. 2h. The authors need to show absolute RPKM/FPKM value of *Kitl* (and also *Cxcl12*) so that readers can form their own opinion on this important issue.

We thank the reviewer for this important suggestion. We have added the RPKM levels of *Kitl* and *Cxcl12* in Supplementary Fig. 2. *Kitl* expression in SECs is extremely low (RPKM: 0.72 ± 0.25). *Kitl* expression in Nestin^{bright} AECs is 98 times higher than SECs and 68 times higher in Nestin^{dim} AECs than SECs. Our qPCR results in Fig. 2i have confirmed the extremely low level of *Kitl* expression in SECs (Relative to GAPDH: 0.0094 ± 0.0064) and the much higher expression in AECs. The reviewer has raised the possibility that SEC may be heterogeneous with a subset expressing higher SCF. However, our FACS analyses of Scf-GFP animals, which have allowed us to evaluate SCF expression at the single cell level, have revealed that SEC were uniformly negative in GFP expression whereas the vast majority of AECs were positive. Taking these data into consideration, we are very confident that the vasculature-derived SCF is from AECs, but not from SECs.

2) I have concerns about the purity of the isolated ECs. If *Epor-Cre* only labels SECs and *Bmx-CreER* only AECs (based on imaging), the FACS data indicate that about 20% to 35% cells do not correspond to real arterial ECs (Fig. 3c and 4d). Moreover, does *Epor-Cre* label other capillary EC subpopulations in the metaphysis, which is not shown in the current data set. What about *Epor-Cre* recombination in other cell populations? Better analysis of the genetic tools and the AEC/SEC populations is necessary.

Heinrich et al. (ref. 20) have reported *Epor-Cre* recombination predominantly in the Ter119-positive fraction, whereas less recombination was seen in Ter119-negative BM and spleen cells. This seems inconsistent with the current study.

We agree that, as reported by Heinrich et al. (ref. 20), *Epor-Cre* induces Cre recombination in the Ter119⁺ erythroid progenitor fraction. We have found that it labels about 79% of MEP and a very low percentage of HSPC and mature cells (data added to Supplementary Fig. 6). These progenitors are not visible in our immunofluorescence images because the level of iTdTomato fluorescence in erythroid progenitors is much lower than in endothelial cells and would require longer exposure to be detected.

After the submission of our manuscript (and mentioned by Reviewer 1), a recent study has suggested that bone marrow endothelial cells express *Epor* as determined by qPCR (Minato et al. *Cell Structure and Function*, 2017). However, the differential

expression of *Epor* between AECs or SECs was not demonstrated. Our data show that *Epor* is largely expressed on SECs by qPCR, RNAseq, immunofluorescence, and FACS analyses using a reporter line (Fig. 2g, Fig. 3 a-c, and Supplementary Fig. 5a). Analysis of other tissues also show expression in other endothelia. Our immunofluorescence analyses are consistent with the FACS data in that a small portion of AECs are targeted by *Epor-Cre* (please see upper left in Fig. 3a), but the majority of SECs, suggesting that it is a valid tool to assess the contribution of SECs. We added the new imaging data of metaphysis of *Epor-Cre*; iTdTomato mice in Supplementary Fig. 5b showing that *Epor-Cre* does not label other capillary EC subpopulations in the metaphysis.

3) *Epor-Cre* shows recombination in about 20% of AECs and therefore it is important to show that arterial SCF is not altered in the *EporCre*-induced SCF mutants. Conversely, the authors should show that SCF expression in SECs is not altered in the *Bmx-CreER*-generated mutants.

Along the same lines, SCF-GFP and SCF (*Kitl*) transcripts are prominently expressed in perivascular *Lepr*⁺ or CAR cells (Lei Ding et al., Nature 2012; Noboru Asada et al., NCB 2017). Is perivascular SCF expression altered in the EC-specific mutant models?

We have carried out new experiments in which we sorted the AECs, SECs, and MSCs from *Epor-Cre*; *Scf*^{fl/-} and control mice, quantified the *Scf* transcript in these cells, and found that *Scf* was not altered in AECs or MSCs (identified as CD45⁻ Ter119⁻ CD31⁻ CD51⁺ PDGFR α ⁺) (Supplementary Fig. 7a). In addition, we have sorted the AECs, SECs, and MSCs from *Bmx-Cre*^{ERT2}; *Scf*^{fl/-} and control mice, quantified the *Scf* transcript in these cells, and found that *Scf* is only reduced in AECs, but not SECs or MSCs (Supplementary Fig. 9c).

4) The authors argue that SCF from sinusoidal ECs is dispensable for HSC maintenance under physiological conditions. Does AEC or SEC-derived SCF control BM regeneration after irradiation or chemotherapy? The authors should use *Epor-Cre* and *Bmx-CreER*-specific SCF mutant mice to address this important question.

We thank the reviewer for this suggestion. We have carried out further experiments of bone marrow regeneration after 5-FU injection. We found that SCF derived from AECs also promotes HSC regeneration after 5-FU injection. These new data were added in Figure 6.

5) The FACS plots on the top and bottom right in Fig.1e are identical or almost identical (see dot distribution) even though different antibodies were used. The authors need to check the original data and correct this figure. In Supplementary Fig. 2c and 2d, the overlay plots are not clearly described in the legends. What are the gating areas and how to red/blue dots correspond to each other? Please provide information on how the overlay picture was generated.

We apologize for the confusion. The top and bottom plots of Fig. 1e are similar but not identical. They were obtained from the same mouse injected with both isolectin-AF488 and anti-VEcadherin to show that endothelial cells marked by either method can be separated into AECs and SECs using Sca-1/PDPN. Hence, the cells in the top and bottom panel are indeed the same cells labelled with a different marker, which is why the plots look the same in Fig. 1e right panel, but not in the middle and left panels. With regards to Supplementary Fig. 2c and d (now Supplementary Fig. 3b), we have simplified the panels to show the cells of the AEC gate as **red dots**, and the cells of the SEC gate as **blue dots** on the left, and plotted these cells as overlay on the right.

6) Several studies, including some by the lead author of this manuscript, have shown that arterioles maintain quiescent HSCs. Is this population specifically altered in the *Bmx-CreER*-generated SCF mutant?

We have performed cell cycle analysis in the control and *Bmx-Cre^{ERT2}; Scf^{fl/-}* mice and did not see a significant difference in the percentage of quiescent HSCs (Supplementary Fig. 9f). In addition, we evaluated HSC localization by immunofluorescence and did not see a significant difference in the relationship with arterioles (Supplementary Fig. 9g).

Likewise, HSC numbers are changed in *Bmx-CreERT SCF^{fl/-}* mutants but, strangely, this has no effect on cellularity and differentiation (Suppl. Fig. 4). Please explain.

Our results are consistent with data from the Morrison laboratory who have previously described that the cellularity did not change after SCF deletion using *Tie2-cre* mice (Ding et al *Nature* 2012). This may be due to the fact that cKit-expressing HSC and progenitors represent a minor population of bone marrow cells and endothelial cells are not the only source of SCF in bone marrow. The effect of HSC may thus reflect both their increased dependence on the cytokine (compared to more differentiated progenitors) and their location in the bone marrow.

7) By combining Sca-1 and PDPN with CD45/Ter119/CD31, the authors separated AECs and SECs, which is fundamental for this work. In Fig. 1f, PDPN (i.v.) is indicated. But in the main text part related to this figure (line 89-95), PDPN is done by ex vivo staining, not intravenous labeling. Please make sure the correct experimental procedure is described and explain whether there could be differences between in ex vivo staining and intravenous labeling. The authors also need to provide more detail in the Methods (e.g. dosage, antibody catalog numbers, treatment duration).

We apologize for the confusion. Anti-PDPN was given i.v. throughout this manuscript. In the text, the antibody which was used ex vivo was anti-CD31, not anti-PDPN. “These results confirm that the ex vivo-stained CD31⁺ fraction contains non-endothelial cells and suggest that PDPN and Sca-1 can be used to study prospectively purified populations of arterial and sinusoidal ECs (henceforth called AECs and SECs, respectively).” We thank the reviewer for the suggestion and added more detailed information in the methods.

8) In Fig. 2a, both AECs and SECs are Nestin-GFPdim. However, it is difficult to see GFPdim signal by immunostaining. Most of appear to be GFP negative (Fig. 1d, 3a, 4a). Do Nestin-GFPdim (GFP+) ECs only represent a specific population of sinusoidal ECs? Please show Nestin-GFP dim staining in SECs in Fig. 1d, Fig. 3a and Fig. 4a.

The reviewer is correct that Nestin-GFPdim is not readily visible by immunofluorescence microscopy and overshadowed by arterial-associated cells that appear very bright. SECs appear uniformly Nestin-GFPdim whereas AECs are heterogeneous in Nestin-GFP expression. This is the reason why we have sorted them separately for RNA-seq analyses. However, the analysis shows that both AEC subsets have the same arterial profile and similar expression of HSC niche factors.

Nestin-GFP also labels perivascular cells that are closely associated with ECs. The authors need to provide images with higher magnification (40x or 63x) to show co-localization in Fig. 1d, 3b and 4b.

The reviewer is correct that Nestin-GFP also labels perivascular cells that are closely associated with ECs. However, we are not able to reliably separate them by imaging, even with higher magnification. On the other hand, we were able to show by FACS that *Bmx-Cre*^{ER} labels AECs and *Epor-Cre* labels SECs in Fig. 3c and 4d. For higher magnification of Nestin-GFP imaging around arteries, please see Fig. 1a, 4a, 4b, 4i in our recent publication (Asada et.al Nature Cell Biology, 2017) and Fig. 2b, 3a in another publication (Kunisaki et al Nature 2013).

Previous study from the authors showed Nestin-GFP positive cells are CD31-negative and CD34, VE-cadherin-negative (Mendez-Ferrer et. al., Nature 2010). The current study shows that Nestin-GFP cells are positive for CD31 and VE-Cad. Please explain.

In our initial analyses of Nestin-GFP stromal cells (Mendez-Ferrer et al. Nature, 2010), trypsin was used to dissociate stromal cells of the bone marrow. We have subsequently found that trypsin altered some surface proteins, including CD31, from cell surface, thereby preventing antibody binding. The use collagenase 4 and dispase for bone marrow cell dissociation does not alter endothelial cell antigens and shows that endothelial cells indeed express dim levels of Nestin-GFP.

Minor points:

9) Something is wrong in the text referring to Supplemental Fig. 2b. Please check and correct.

We thank the reviewer and have made the correction.

10) In Fig. 2k, please explain the meaning of the label "PE" on the vertical axis In Fig. 2l, the label for the vertical axis and the percentage of gated cells are missing.

We thank the reviewer for bringing this to our attention. We have replaced the dot plot by histogram plots in Fig. 2k and included the wild-type mice as control.

11) In Fig. 4d, please show quantitative data for the percentage of AECs in Bmx-CreER+ cells.

The percentage was added.

12) In Fig. 5i, after irradiation, the percentage of Bmx-CreER+ cells in AECs decreased at 4 months (21.7%) relative to 1 month (42.9%). But in Fig. 5g, Bmx-CreER+ cells seem increased at 4 months. Please explain.

We thank the reviewer for the comment. In Fig 5g (4 months), the sternum was cut near the bone surface which in general has a higher density of arterioles. We have updated Fig. 5g with a more representative image.

In Fig. 5i, percentage of AECs/SECs is not indicated in the plots.

The percentage was added.

Reviewer #3 (Remarks to the Author):

Overall the manuscript is well written and the conclusions supported by the data. The manuscript is of great interest to the field and will substantially increase the understanding of the different types of vasculature within the BM.

We thank the reviewers for the positive comments on our manuscript.

However, some of the language of the manuscript is very definite, yet this is not proven by the data and should be toned down. For example, the authors commonly say AEC or SEC specific genes, yet all the data represents relative expression in one population to another, not that the gene is only expressed in this population and is not expressed in any other cell type in the body. As the latter is highly unlikely and has been proven to be untrue for many previously reported “specific genes”, less definitive terminology would be appropriate.

We agree with the reviewer. Most genes are detectable in both populations. Accordingly, we changed the term to “AEC-enriched genes” and “SEC-enriched genes” to better reflect our findings.

In addition, no isotypes or other controls are included for any of the FACS analysis or immunohistochemistry. These should be included.

We thank the reviewer for the comment. We have updated the manuscript as detailed below:

- 1) Immunofluorescence staining of an isotype-matched control Ab was included in our experiments for anti-PDPN and is now shown in Supplementary Fig. 1b.
- 2) For the *Scf*-GFP and *Cxcl12*-GFP reporter mice analysis, we added wild-type mouse data as control and updated the Fig. 2k and 2l.
- 3) An isotype-matched control for anti-CD45/Ter119 (both are of the same isotype) is shown below in Reviewer Fig. 2.

Importantly, the authors describe the populations as CD45-Ter119, yet in the vast majority of the dotplots in the manuscript showing CD45ter119 versus CD31, a real proportion (ore even most) of the population labelled as CD45Ter119- would have to be considered CD45Ter119dim. An isotype and an unlabeled control would be very helpful here, but the authors should discuss this.

We thank the reviewer for pointing this out. When staining enzymatically digested bone marrow for CD45 and Ter119 to isolate the non-hematopoietic fraction, we typically see a smear without a clear separation of the negative population. If one plots CD45/Ter119 and CD31, separate populations emerge. Reviewer Fig. 2 below shows a representative staining: the left panel was staining an isotype-matched control with anti-CD31, and the right panel was stained with anti-CD45/Ter119 with anti-CD31. It is clear that the CD31+ population is in the CD45/Ter119 negative gate.

For Reviewer Fig. 2:

For Reviewer Fig. 2

Enzymatically digested bone marrow cells were stained with an isotype-matched control with anti-CD31(left) or anti-CD45/Ter119 with anti-CD31(right).

Minor points:

Line 119 and 120. The authors contradict themselves. In line 119 they state sele, selp etc. are more highly expressed in SEC than AEC, then in line 120, they reverse this and say higher in AEC than SEC. The data in sup 2b supports the statement in line 119.

We thank the reviewer for noticing the error which was corrected. Adhesion molecules are higher on SECs than AECs.

REVIEWERS' COMMENTS:

Reviewer #1 (Remarks to the Author):

The authors have addressed most of my concerns and the revised version is much improved.

Reviewer #2 (Remarks to the Author):

The revision has fully addressed all my questions and concerns. In my view, the manuscript is now suitable for publication.

Reviewer #3 (Remarks to the Author):

The authors have now satisfactorily addressed my concerns.

REVIEWERS' COMMENTS:

Reviewer #1 (Remarks to the Author):

The authors have addressed most of my concerns and the revised version is much improved.

Reviewer #2 (Remarks to the Author):

The revision has fully addressed all my questions and concerns. In my view, the manuscript is now suitable for publication.

Reviewer #3 (Remarks to the Author):

The authors have now satisfactorily addressed my concerns.

We thank all reviewers for the favorable comments.